# ATRX restricts Human Cytomegalovirus (HCMV) viral DNA replication through heterochromatinization and minimizes unpackaged viral genomes

**Ryan M. Walter, Kinjal Majumder, Robert F. Kalejta** *

Institute for Molecular Virology and McArdle Laboratory for Cancer Research, University of Wisconsin-Madison, Madison, Wisconsin, United States of America

* rfkalejta@wisc.edu

**Data Availability Statement:** ATAC-seq data files have been deposited in the NCBI Gene Expression Omnibus database (https://www.ncbi.nlm.nih.gov/

## Abstract

ATRX limits the accumulation of human cytomegalovirus (HCMV) Immediate Early (IE) proteins at the start of productive, lytic infections, and thus is a part of the cell-intrinsic defenses against infecting viruses. ATRX is a chromatin remodeler and a component of a histone chaperone complex. Therefore, we hypothesized ATRX would inhibit the transcription of HCMV IE genes by increasing viral genome heterochromatinization and decreasing its accessibility. To test this hypothesis, we quantitated viral transcription and genome structure in cells replete with or depleted of ATRX. We found ATRX did indeed limit viral IE transcription, increase viral genome chromatinization, and decrease viral genome accessibility. The inhibitory effects of ATRX extended to Early (E) and Late (L) viral protein accumulation, viral DNA replication, and progeny virion output. However, we found the negative effects of ATRX on HCMV viral DNA replication were independent of its effects on viral IE and E protein accumulation but correlated with viral genome heterochromatinization. Interestingly, the increased number of viral genomes synthesized in ATRX-depleted cells were not efficiently packaged, indicating the ATRX-mediated restriction to HCMV viral DNA replication may benefit productive infection by increasing viral fitness. Our work mechanistically describes the antiviral function of ATRX and introduces a novel, pro-viral role for this protein, perhaps explaining why, unlike during infections with other herpesviruses, it is not directly targeted by a viral countermeasure in HCMV infected cells.

## Author summary

Viruses infect host cells and often kill them, so cells encode proteins geared to protect them from viral predators. Viruses, as obligate intracellular parasites, rely on host cell proteins for functions they require to replicate. Thus, cellular proteins can be anti-viral or pro-viral. The cellular ATRX protein is clearly antiviral for Human Cytomegalovirus (HCMV) and other viruses because it inhibits viral protein accumulation and activates cellular antiviral transcription. Surprisingly, we found that ATRX also plays a pro-viral

geo/) under the series accession number GSE262839.

**Funding:** This work was supported by a grant from the National Institutes of Health (NIH) to RFK (AI130089). RMW was supported by NIH training grant T32 CA009135 (to William M. Sugden). The funders had no role in study design, data collection and analysis, decision to publish, or preparation of the manuscript.

**Competing interests:** The authors have declared that no competing interests exist.

role by limiting viral DNA accumulation to allow for efficient genome packaging into capsids and infectious virions. Our work highlights how the same cellular protein can be antiviral for one aspect of infection but pro-viral for a different step.

## Introduction

The Alpha Thalassemia/Mental Retardation factor (ATRX) protein is found at Pro-Myelocytic Leukemia protein Nuclear Bodies (PML-NBs). PML-NB resident proteins impair aspects of productive (lytic) herpesvirus replication while supporting the ability of these viruses to establish and maintain latency [1]. ATRX levels are decreased in Herpes Simplex Virus Type 1 (HSV-1) lytically infected cells by the viral ICP0 Immediate Early (IE) protein that activates viral Early (E) gene transcription [2–4]. HSV-1 also encodes miRNAs that target the ATRX transcript, and thus ATRX levels remain low throughout HSV-1 lytic infections [3]. Other herpesviruses, such as Human Cytomegalovirus (HCMV) and Epstein Barr Virus (EBV) do not target ATRX but instead encode tegument (virion structural) proteins that associate with its binding partner, the Death Domain Associated Protein (Daxx) and activate viral IE transcription. EBV BNRF1 disrupts Daxx-ATRX complexes [5,6], and HCMV pp71 induces the proteasome-dependent, ubiquitin-independent degradation of Daxx [7,8].

When associated with ATRX, Daxx deposits histone H3.3 trimethylated at lysine 9 (H3K9me3) onto DNA [9]. Herpesvirus double-stranded DNA genomes are devoid of histones when they first enter the nucleus, but then become rapidly associated with cellular histones to form viral chromatin [10–16]. Octamers of histones (nucleosomes) wrap DNA around their outer surface and increase the compaction of a DNA strand [17]. Post-translational modification of histone tails influences their ability to compact DNA and therefore impacts a genome's transcriptional potential [18]. For example, the H3K9me3 epigenetic modification found on histones deposited by Daxx is transcriptionally repressive at promoters [9,19,20]. Multiple cellular proteins can deposit histones onto DNA, but the only one known to deposit histones onto HCMV DNA is Daxx.

We have shown that Daxx deposits H3.3 onto the HCMV Major Immediate Early Promoter (MIEP) at the start of both lytic and latent infections, leading to transcriptional repression [21]. At the start of a latent infection of incompletely differentiated myeloid cells, we showed that virion-delivered pp71 remains in the cytoplasm, Daxx remains stable and deposits histones onto latent viral genomes to repress viral lytic phase transcription [21–23]. However, at the start of lytic infection of fibroblast cells, we showed tegument delivered pp71 travels to the nucleus, induces Daxx degradation, and thereby stimulates lytic phase viral transcription [7]. At later times during lytic HCMV infection, the Daxx protein reaccumulates [7,24].

pp71 also displaces ATRX from its localization at PML-NBs during HCMV lytic infection [25]. Recombinant HCMVs lacking the gene (UL82) encoding for pp71 show decreased viral major IE transcript and protein accumulation upon lytic infection of fibroblasts [26], and knockdown of ATRX [25] increased viral IE protein accumulation from a pp71-null virus. ATRX knockdown cells also showed an increase in IE1-positive cells during a Wild Type (WT) HCMV lytic infection [27]. Conversely, ATRX overexpression decreased IE protein accumulation during WT HCMV lytic infection [28]. Neither the impact of ATRX on HCMV IE transcription nor the molecular mechanisms through which ATRX exerts its effects on HCMV IE protein levels have been explored.

In addition to helping Daxx deposit histone H3.3 to modulate transcription, the ATRX/Daxx complex also regulates DNA replication, repair, and telomere integrity [29–37].

Specifically, ATRX limits the formation of secondary structures (G4 quadruplexes and R-Loops) that cause DNA polymerase pausing and replication stress [38–41]. ATRX also has Daxx-independent functions, including the negative regulation of macroH2A deposition at the HBA1 locus [42], X-chromosome inactivation via the PRC2 complex [43,44], and promoting the IRF-3 dependent expression of a subset of ISGs involved in DNA replication and cytokine-mediated signaling [45].

Here we show that ATRX decreases the accessibility of HCMV genomes, participates in their chromatinization and heterochromatinization, and exerts its known impact on viral IE gene expression at the level of transcription. Our work mechanistically defines the anti-viral effects of ATRX at the start of HCMV lytic infections. We further show that ATRX decreases viral DNA replication independent of its effect on IE and E gene expression. However, we also show here that ATRX restriction of viral DNA replication acts in a pro-viral manner to suppress the overproduction of viral genomes that would exceed packaging capabilities. Thus, in stark contrast to its anti-viral function at the start of lytic infection, ATRX seems to act in a pro-viral manner at later times, perhaps explaining why HCMV (unlike HSV-1 and KSHV) does not directly target ATRX [3,46] and why its binding partner Daxx is allowed to reaccumulate after initial degradation in cells lytically infected with HCMV.

## Results

### Depletion of ATRX increases the accumulation of IE transcripts early during lytic infection

We initially investigated the impact of ATRX on HCMV IE transcription using both a constitutive knockdown, as well as a genetic deletion (knockout). To generate constitutive knockdown cells, we transduced immortalized human foreskin fibroblasts (HFFs) with a pLKO.1-puro lentivirus expressing a scrambled (SCR) shRNA or one targeting ATRX [47]. Drug-resistant populations displayed decreased ATRX protein levels in the presence of the targeting shRNA (ATRX-KD) compared to the control (Fig 1A). ATRX levels in the knockdown cells were reduced by 80% (Fig 1B). Daxx levels were not significantly impacted (Fig 1A and 1B). To knockout ATRX, we transfected HFFs with Cas9 mRNA and a single guide RNA (sgRNA) targeting the ADD domain of ATRX (Fig 1C). Sequencing the population determined that ~25% of the cells had insertions or deletions (INDELs) (Fig 1D) in the single allele of ATRX (it is found on the X chromosome) in these (male) cells. However, following clonal expansion of single cells, only 1% of clones (2 out of 200) contained an INDEL (Fig 1E), and both lesions were a 3bp deletion that removed an amino acid (R250) but left the rest of the protein in frame. The clone we examined (AKO) had reduced ATRX levels compared to parental (WT) cells, but clearly continued to express the protein as compared to ATRX-null U-2 OS cells [48] (Fig 1F). Thus, our knockout attempts generated a hypomorph, similar to previous attempts at ATRX knockout in these cells [4], suggesting that ATRX is an essential gene in HFFs. Both the ATRX hypomorph and knockdown showed similar increases in IE (IE1 and IE2) transcript accumulation after infection with HCMV strain AD169 at an MOI of 0.1 (Fig 1G), as expected from previous observations of increased IE protein accumulation after ATRX knockdown [25,27]. We chose an MOI of 0.1 because previous studies have demonstrated the effects of intrinsic defense proteins such as ATRX are readily saturated at high MOIs [26,49–51], and confluent fibroblasts because this culture condition best approximates the in vivo setting [52]. Because of the variability we observed in the hypomorph cells, we confined our subsequent analyses to the constitutive knockdowns.

ATRX knockdown also increased IE transcription from the TB40/E strain of HCMV (Fig 2A). Because pp71 inactivates the Daxx-ATRX complex, we hypothesized ATRX knockdown

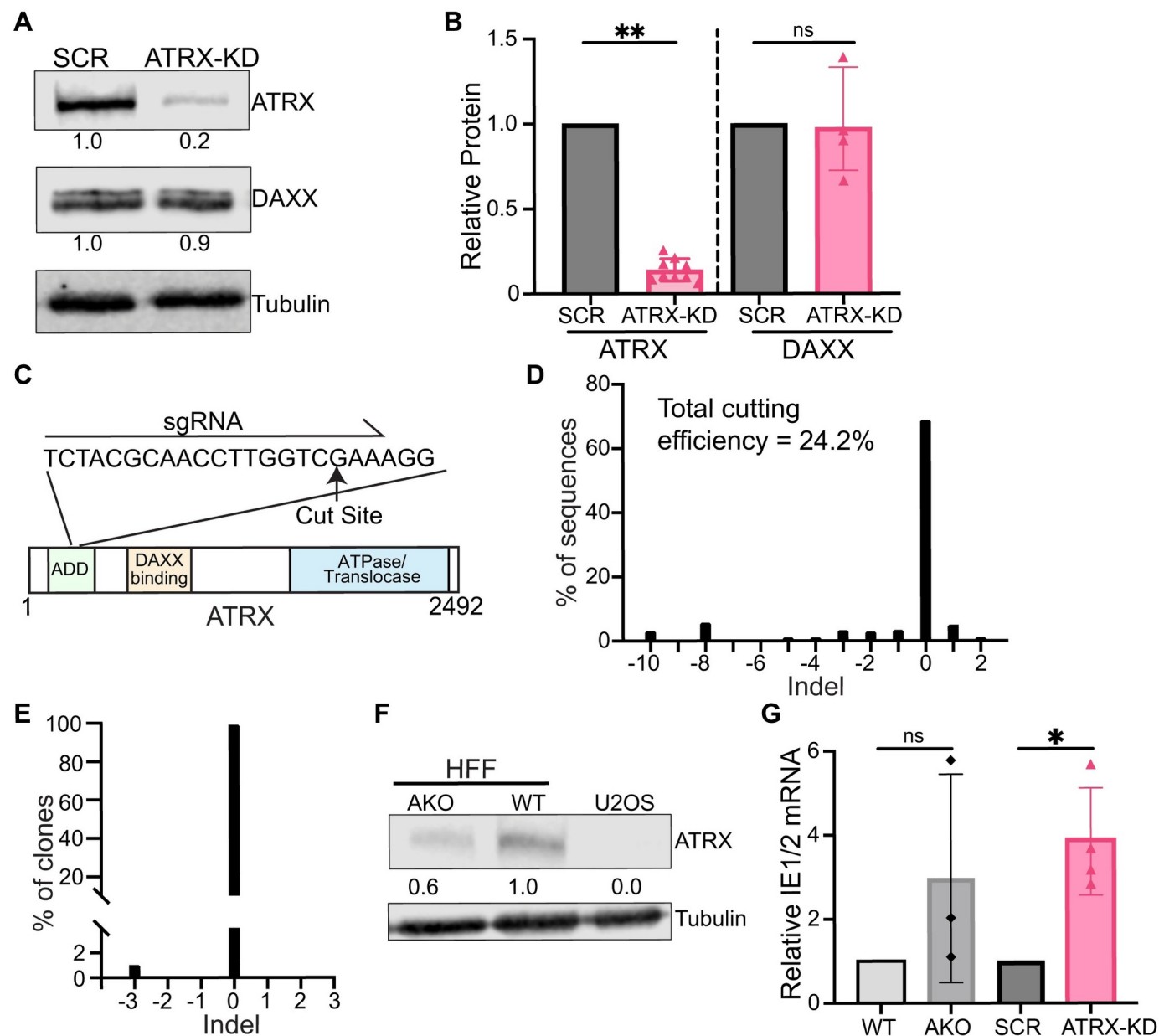

**Fig 1. Depletion of ATRX in fibroblasts increases IE transcript accumulation during HCMV lytic infections. A)** Representative western blot of human foreskin fibroblasts (HFF) transduced with pLKO lentivirus expressing either a scrambled (SCR) shRNA control sequence or an shRNA sequence targeting ATRX (ATRX-KD). Values represent protein levels normalized to tubulin and relative to SCR. **B)** Quantification of multiple biological replicates of the experiment in panel A. ATRX and DAXX signals were normalized to tubulin and are reported relative to SCR. **C)** Schematic for targeting of sgRNA to the ATRX coding sequence to generate ATRX knockout cells. **D)** TIDE analysis from the polyclonal population of sgRNA transfected HFF cells. Total cutting efficiency is the percentage of Sanger sequencing signal with predicted indels at the Cas9 cut site. **E)** Clonal populations of cells (n = 200) were screened by Sanger sequencing for indels and plotted for the percentage of cells with indicated indels at the Cas9 cut site. **F)** Western blot for ATRX protein in wild-type (WT) positive control HFF cells, ATRX knockout cells (AKO), and negative control U-2 OS cells. Values represent ATRX protein level normalized to tubulin and relative to HFF-WT. **G)** HFF cells were infected with AD169 at an MOI of 0.1 and total RNA was harvested at 3 hpi, quantified by RT-qPCR, and normalized to GAPDH. Plotted values are relative to respective control cells. All experiments were performed with a minimum of 3 biological replicates. Error bars represent standard deviation and statistically significant differences are indicated with asterisks (* = P<0.05, ** = P<0.01, *** = P<0.001; Wilcoxon rank sum).

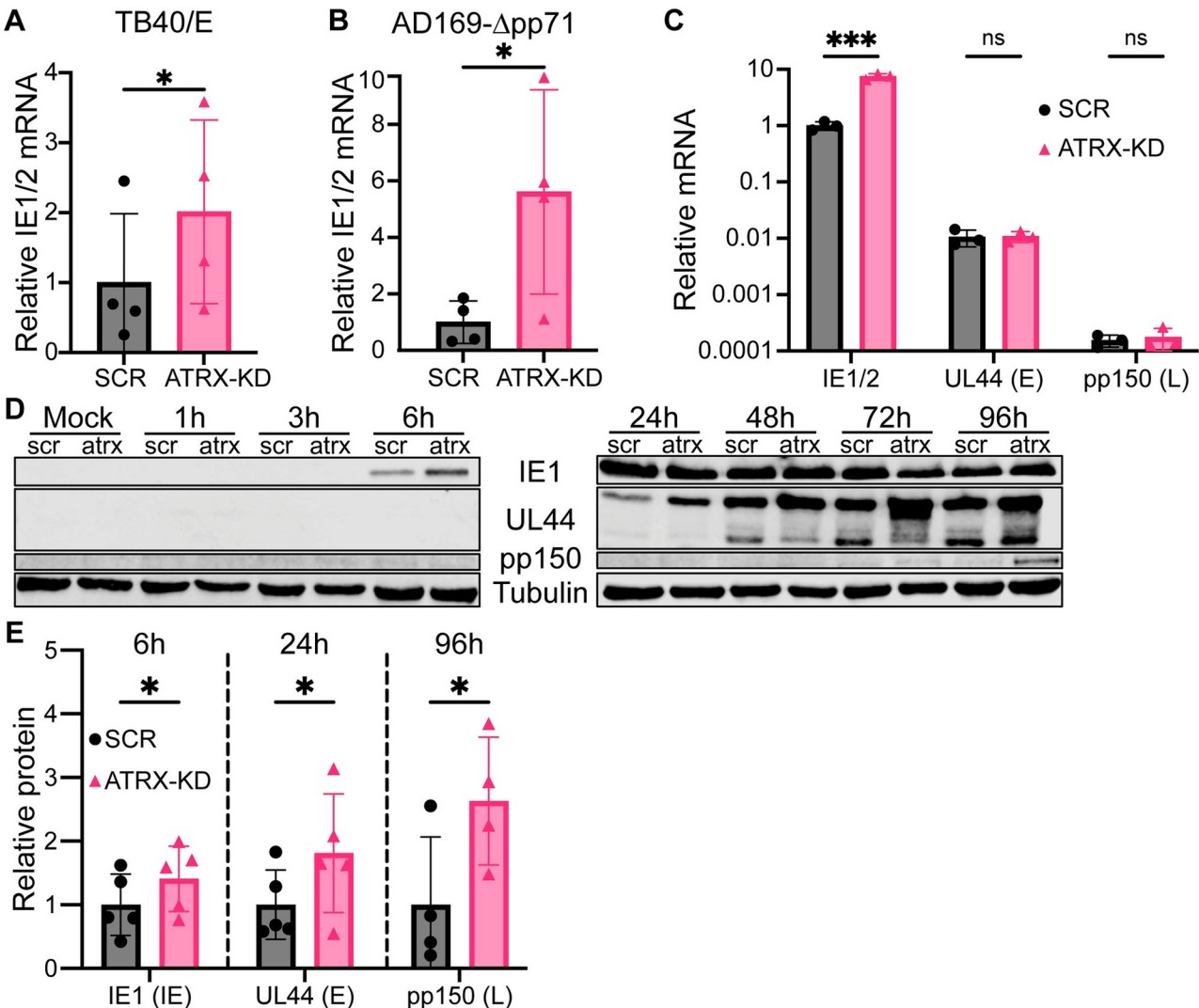

**Fig 2. ATRX knockdown increases IE transcript and Early and Late protein accumulation during HCMV lytic infections. A)** HFF cells were infected with TB40/E at an MOI of 0.1 and total RNA was harvested at 3 hpi, quantified by RT-qPCR, normalized to GAPDH, and reported relative to SCR. **B)** Experiments performed as in panel A with a UL82-null strain of HCMV (AD169-Δpp71). **C)** HFF cells were infected with AD169 at an MOI of 0.1 and quantified by RT-qPCR at 3 hpi. Reported values are normalized to GAPDH and relative to SCR-IE1/2. **D)** HFF SCR (scr) and ATRX-KD (atrx) cells were infected with AD169 at an MOI of 0.1, harvested at the indicated time points, and cell lysates were analyzed by western blotting. Representative images are shown. **E)** Quantification of blots from experiments shown in panel D for IE1, UL44, and pp150 at 6, 24, and 96 hpi respectively. Reported values are normalized to tubulin and plotted relative to SCR. All experiments were performed with a minimum of 3 biological replicates. Error bars represent standard deviation and statistically significant differences are indicated with asterisks (* = P<0.05, ** = P<0.01, *** = P<0.001; t-test).

would increase IE transcription from a pp71-null virus to a greater extent than WT virus. Indeed, we observed an ~4-fold increase in IE transcripts for WT HCMV compared to control cells, but a >6-fold increase from a pp71-null virus (Fig 2B). ATRX knockdown failed to increase transcription of a viral Early (UL44) or Late (UL32/pp150) gene at 3hpi (Fig 2C), indicating the canonical kinetic cascade of lytic gene transcription was not disrupted by ATRX depletion. However, when we extended our time course to later stages of infection, we saw increases in IE1, UL44 (Early, 24hpi), and pp150 (Late, 96hpi) protein accumulation in

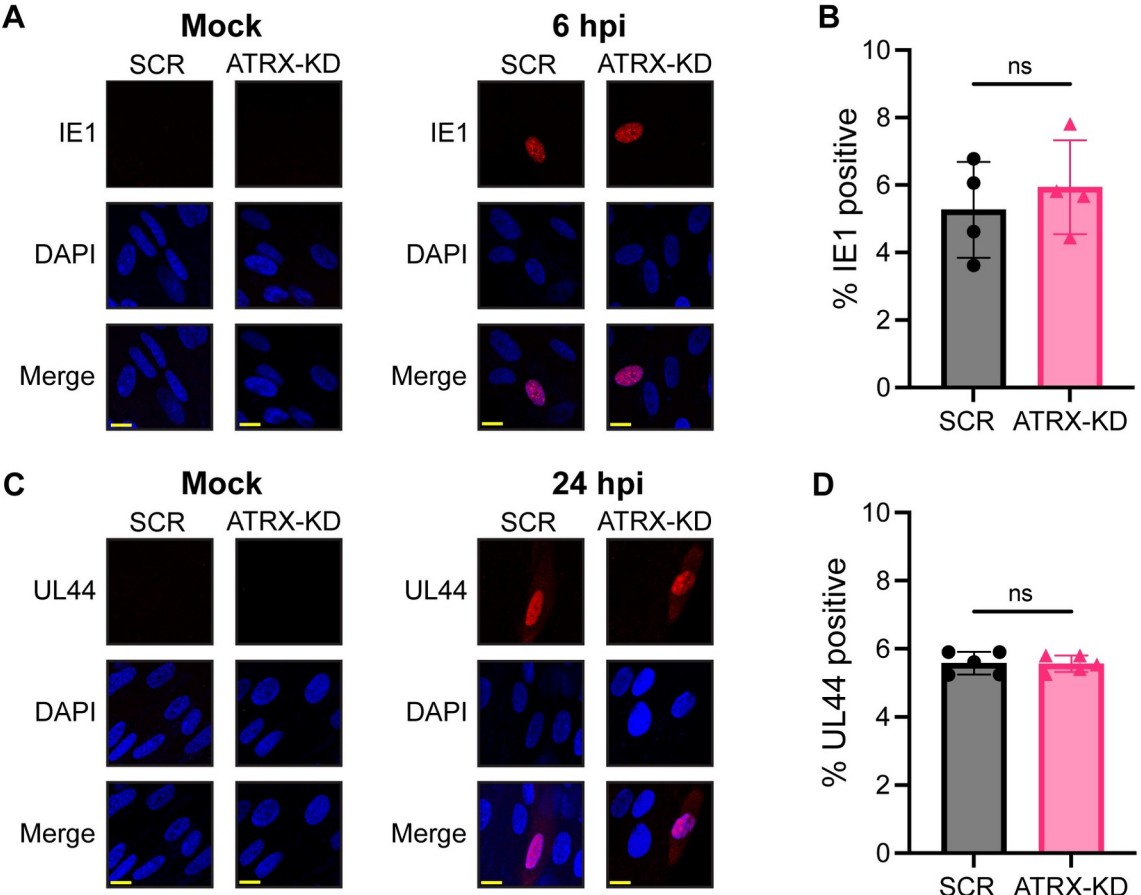

**Fig 3. ATRX knockdown does not affect the number of cells initiating lytic viral IE and E gene expression. A)** The indicated HFF cells were mock infected (left) or infected with AD169 at an MOI of 0.1 (right), fixed at 6 hpi, and analyzed by immunofluorescence for IE1. **B)** The percentage of cells positive for IE1 at 6 hpi. **C)** The indicated HFF cells were mock infected (left) or infected with AD169 at an MOI of 0.1 (right), fixed at 24 hpi, and analyzed by immunofluorescence for UL44. **D)** The percentage of cells positive for UL44 at 24 hpi. Representative images of the indicated viral protein (red) and DAPI-stained nuclei (blue) are shown. Yellow scale bars represent 10μm. Experiments were performed with a minimum of 3 biological replicates. Each replicate represents a minimum of 100 cells. Error bars represent standard deviation and statistically significant differences are indicated with asterisks (* = P<0.05, ** = P<0.01, *** = P<0.001; t-test).

ATRK-KD cells compared to controls (Fig 2D and 2E). This experiment shows that ATRX-KD increases viral protein accumulation but cannot determine if ATRX knockdown cells were accumulating more viral proteins per cell, or if more cells were initiating infection. To address this question, we used immunofluorescence (IF) to visualize the number of individual cells expressing viral proteins. At 6hpi, there was no statistically significant difference in the percentage of cells expressing IE1 in ATRX-KD cells compared to SCR (Fig 3A and 3B). The same was true for the levels of UL44 at 24hpi (Fig 3C and 3D). We conclude the same number of SCR and ATRX-KD cells are generating HCMV proteins, but that ATRX-KD cells are producing more viral proteins per cell than are SCR cells. These experiments cannot differentiate between direct effects of ATRX on UL44 gene expression or protein levels increasing indirectly downstream of greater IE1 production. However, we can conclude that ATRX suppresses the level of HCMV IE transcription.

## ATRX promotes heterochromatinization and reduces the accessibility of HCMV genomes

The Daxx/ATRX complex deposits H3K9me3-marked histone H3.3 onto DNA. Because ATRX knockdown promotes HCMV IE transcription, we hypothesized that ATRX restricts the accessibility of the HCMV MIEP by generating heterochromatin at that locus. We tested this hypothesis with three independent techniques, Formaldehyde Assisted Isolation of Regulatory Elements (FAIRE), the Assay for Transposase Accessible Chromatin combined with sequencing (ATAC-seq), and Chromatin-Immunoprecipitation (ChIP). Using FAIRE-qPCR assays, we found the MIEP was ~3-fold more accessible in ATRX knockdown cells than in control cells (Fig 4A). The accessibility of a cellular positive control (Actin-B promoter), and the inaccessibility of a cellular negative control (a heterochromatin region of chromosome 12) confirmed the accuracy of our assay (Fig 4B) and showed no difference between control and ATRX knockdown cells.

We next extended our accessibility analysis to the entire viral genome with ATAC-seq. Control and ATRX knockdown cells were infected, and input viral DNA was quantitated by qPCR to confirm equal delivery of viral genomes (Fig 4C). The HCMV genome constituted a higher percentage of total reads in ATRX knockdown cells compared to control cells (Fig 4D), and the viral genome showed broadly higher normalized read counts in ATRX-KD cells (Fig 4E) demonstrating increased accessibility in the absence of ATRX. We conclude ATRX globally restricts the accessibility of the HCMV genome. Interestingly, in the context of the global increase in accessibility, in ATRX-KD cells we detected increased accessibility specifically around the distal enhancer (DE) of the MIEP (Fig 4E) that controls IE1 transcription, which we also found elevated in ATRX-KD cells (Fig 2A–2C). Conversely, the promoters for UL44 or UL32/pp150, genes whose transcription was not enhanced at this timepoint by ATRX knockdown (Fig 2C) did not show increased accessibility in ATRX KD cells (Fig 4E).

The global impact of ATRX knockdown on HCMV genome accessibility contrasts against the seemingly locus-specific impact at the MIEP DE, prompting us to examine the entire HCMV genome for features that might locally concentrate the impact of ATRX. To determine if there were any sequence-specific hotspots for ATRX-mediated restriction of HCMV genome accessibility, we used bigWigAverageOverBed analysis to identify the HCMV genomic regions showing the greatest increases in accessibility in ATRX-KD cells [53], and then analyzed those loci (Table 1). We first used MEME analysis [54] to scan for DNA motifs enriched in the top and bottom 10% of differentially accessible regions of the HCMV genome in ATRX KD cells compared to SCR cells (Fig 4F). Interestingly, the motif for the top differentially accessible regions in the absence of ATRX was not a conserved transcription factor binding site but had a much higher GC content than those showing the lowest increase in accessibility (Fig 4F and 4G; Table 1), consistent with the ability of ATRX to bind GC-rich DNA [38,55]. We plotted the location of the top and bottom 10% differentially accessible regions and found these sites were distributed broadly across the HCMV genome (Fig 4H). Curiously, the MIEP DE is not particularly GC-rich. Therefore, high GC content in viral genomic sequences may control the majority of ATRX-mediated restriction of HCMV genome accessibility, with the possibility of specific sequences also contributing at certain loci. A recent publication identified DNA-sequences that recruit ATRX to open chromatin and found an overrepresentation of motifs that matched the Runt-related transcription factor (RUNX) family, with a RUNX1 motif present in 17% of ATRX binding sites [56]. We scanned the HCMV genome for these RUNX1 motifs and found 59 putative binding sites, including one within the MIEP DE. We plotted the location of these sites across the genome and interestingly RUNX1 sites near the terminal repeat region showed the greatest increases in accessibility in ATRX-KD cells (Fig 4I). Taken

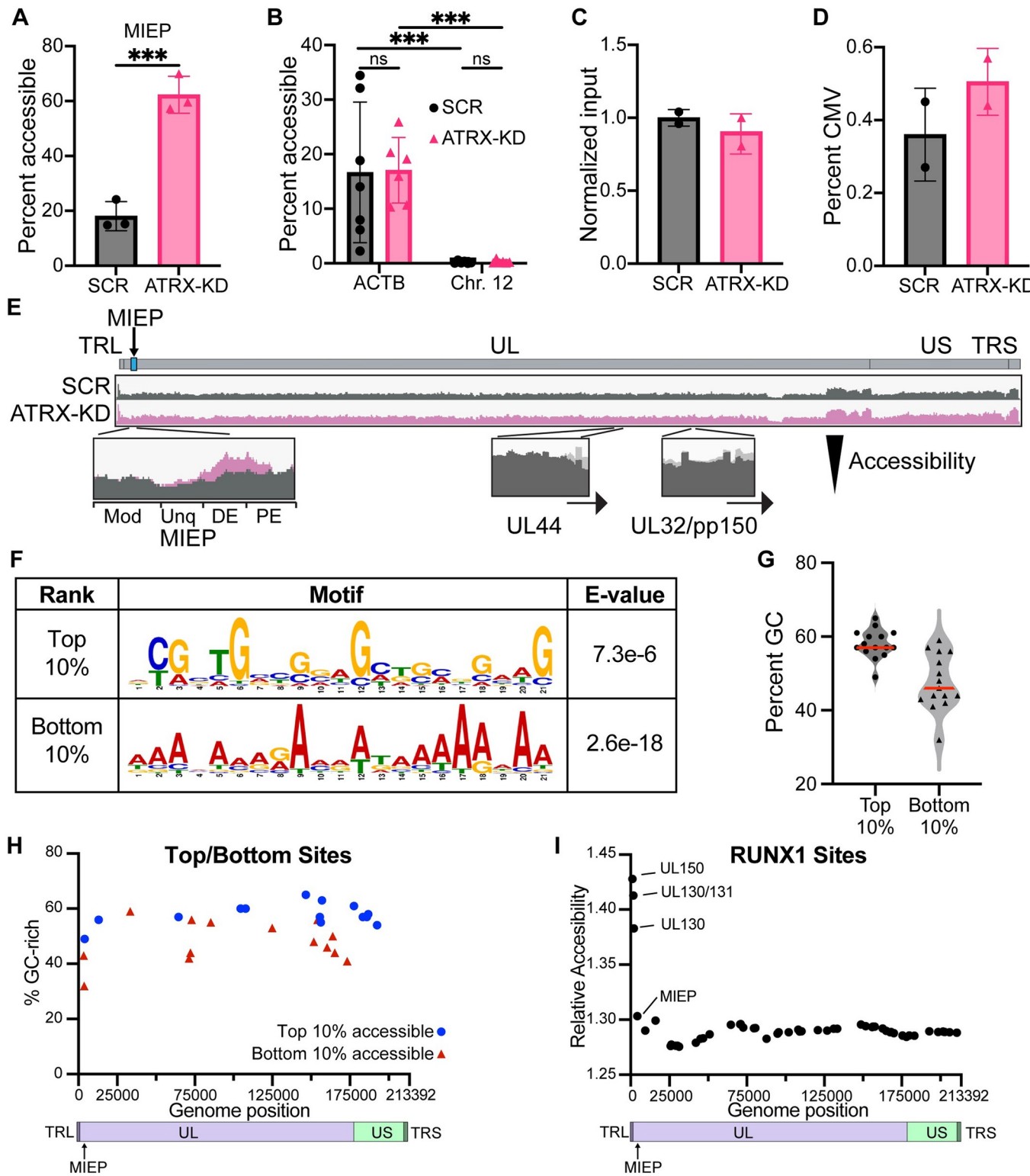

**Fig 4. ATRX knockdown increases the accessibility of infecting HCMV genomes. A)** HFF cells were infected with AD169 at an MOI of 0.1 and harvested at 3 hpi. Accessible DNA was extracted from infected HFF cells by FAIRE and analyzed by qPCR for the HCMV MIEP. Plotted values are normalized to a reversed crosslinked input control sample. **B)** Experiments as in panel A were analyzed for control cellular loci. Experiments were performed with a minimum of 3 biological replicates. Error bars represent standard deviation and statistically significant differences are indicated with asterisks (* = P<0.05, ** = P<0.01, *** = P<0.001; t-test). **C)** HFF cells were infected with AD169 at an MOI of 0.1 and harvested at 3 hpi. Accessible DNA was extracted by ATAC and analyzed by

Illumina sequencing (2x150bp). Input viral genomes were analyzed by qPCR (MIEP), normalized to cellular DNA (Chr.12), and reported relative to SCR. **D)** Percentage of ATAC-seq reads mapping to the HCMV genome relative to reads mapping to the human genome. **E)** Normalized coverage bigwig files from ATAC-seq are displayed using IGV, with zoomed-in coverage on promoter regions. **F)** The HCMV genome was split into 200bp fragments and the top 10% and bottom 10% were analyzed by MEME. Motifs are displayed with their indicated E-values. **G)** G/C content of the top and bottom 10% of the differentially accessible features between SCR and ATRX-KD cells (see also Table 1). **H)** The top (blue) and bottom (red) 10% of sites identified in Table 1 are plotted according to their location on the HCMV genome (x-axis) and their %GC (y-axis). **I)** The 200bp surrounding 59 putative RUNX1 motif sites were plotted according to their position on the HCMV genome (x-axis) and analyzed for their relative accessibility between ATRX-KD and SCR (y-axis).

together our data suggest that ATRX is recruited broadly across the HCMV by the high GC content and potentially to the ends of the genome by RUNX1 motifs.

To determine if ATRX reduced HCMV genome accessibility through heterochromatinization, we used ChIP to quantitate the occupancy of histone H3.3 and the transcriptionally repressive H3K9me3 epigenetic mark. We analyzed two loci, one in the top 10% of loci with increased accessibility (Table 1) and increased transcriptional activity (Figs 1G and 2A–2C) upon ATRX knockdown (the MIEP) and one in the bottom 10% of loci with increased accessibility (Table 1) and that did not show increased transcriptional activity (Fig 5A) upon ATRX knockdown (UL99/pp28). ATRX knockdown decreased H3.3 and H3K9me3 occupancy at the AD169 MIEP (Fig 5B) and decreased the occupancy of total histone H3, H3.3, and H3K9me3 at the MIEP during TB40/E infections (Fig 5C). Conversely, ATRX knockdown did not impact total histone H3, H3.3, or H3K9me3 occupancy at the pp28 gene promoter (Fig 5D). ATRX knockdown also had no impact at a negative control cellular locus (Fig 5E). Taken together, we conclude that ATRX decreases the accessibility of the HCMV MIEP by depositing heterochromatin, thereby repressing transcription.

## ATRX restricts HCMV viral DNA replication initiation and elongation

We next asked whether ATRX affected viral genome replication. We found higher levels of viral genomes in ATRX-KD cells infected with either AD169 (Fig 6A) or TB40/E (Fig 6B) compared to control cells. In these experiments, viral DNA levels are normalized to measured input levels at 3 hpi. Increased viral DNA replication in the absence of ATRX could be because of a direct effect of ATRX on viral DNA replication, or because of an indirect effect downstream of greater IE1 and UL44 protein accumulation (Fig 2D and 2E). To distinguish between these two possibilities, we quantitated viral DNA replication in ATRX-positive and -depleted cells that expressed equal levels of IE1 and UL44 achieved by allowing these proteins to accumulate in the presence of the viral DNA replication inhibitor phosphonoacetic acid (PAA) (Fig 6C). First, we confirmed that PAA treatment did not affect the increase in IE1 and UL44 protein in ATRX-KD cells compared to SCR at 6 and 24 hpi respectively (Fig 6D and 6E).

**Table 1. Differential accessibility of the HCMV genome in ATRX-KD cells.**

| Top 10 | Locus | UL16 | UL15A | US14 | UL17 | US10 | UL78 | RL1 | US13 | MIEP Distal Enhancer | US20 |
|---|---|---|---|---|---|---|---|---|---|---|---|
| | Ratio | 1.587 | 1.585 | 1.507 | 1.500 | 1.450 | 1.421 | 1.417 | 1.415 | 1.402 | 1.395 |
| | % GC | 55 | 63 | 58 | 57 | 57 | 57 | 61 | 57 | 49 | 54 |
| Bottom 10 | Locus | UL99/pp28 | UL22A | UL74 | UL6 | RL6 | UL73 | UL11 | OriLyt | UL19 | MIEP Modulator |
| | Ratio | 1.166 | 1.163 | 1.145 | 1.145 | 1.141 | 1.130 | 1.115 | 1.103 | 1.093 | 0.951 |
| | % GC | 59 | 48 | 42 | 50 | 41 | 44 | 46 | 55 | 56 | 43 |

Normalized ATAC-seq coverage across HCMV loci was calculated with bigWigAverageOverBed [53] and used to determine the relative accessibility (Ratio) in ATRX-KD compared to SCR cells. The average GC richness (% GC) for the top 10 loci showing the most highly increased accessibility in the absence of ATRX was 58%; for the bottom 10 showing the least highly increased accessibility it was 48.4%.

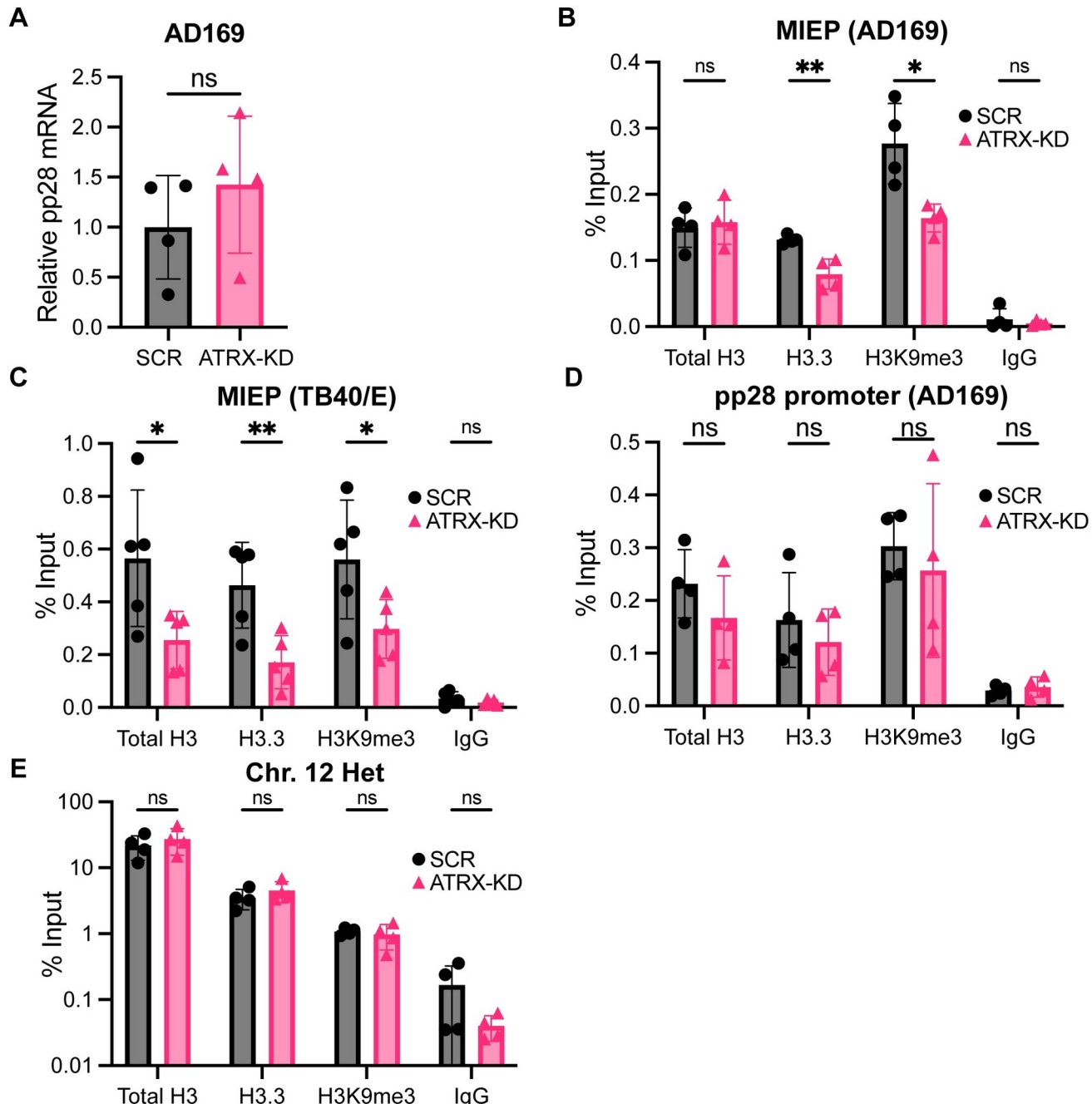

**Fig 5. ATRX-knockdown decreases the deposition of heterochromatin on incoming HCMV genomes. A)** HFF cells were infected with AD169 at an MOI of 0.1 and total RNA was harvested at 3 hpi, quantified by RT-qPCR, normalized to GAPDH, and reported relative to SCR. **B)** HFF cells were infected with AD169 at an MOI of 0.1 and harvested at 3 hpi. ChIP was performed with the indicated antibodies and analyzed by qPCR for the MIEP. Total H3, H3.3, H3K9me3, and IgG are plotted as percent input. **C)** Experiments performed as in panel B with the TB40/E strain of HCMV. **D)** Experiments performed as in panel B for the viral pp28 promoter. **E)** Experiments performed as in panel B for a control cellular locus. All experiments were performed with a minimum of 3 biological replicates. Error bars represent standard deviation and statistically significant differences are indicated with asterisks (* = P<0.05, ** = P<0.01, *** = P<0.001; t-test).

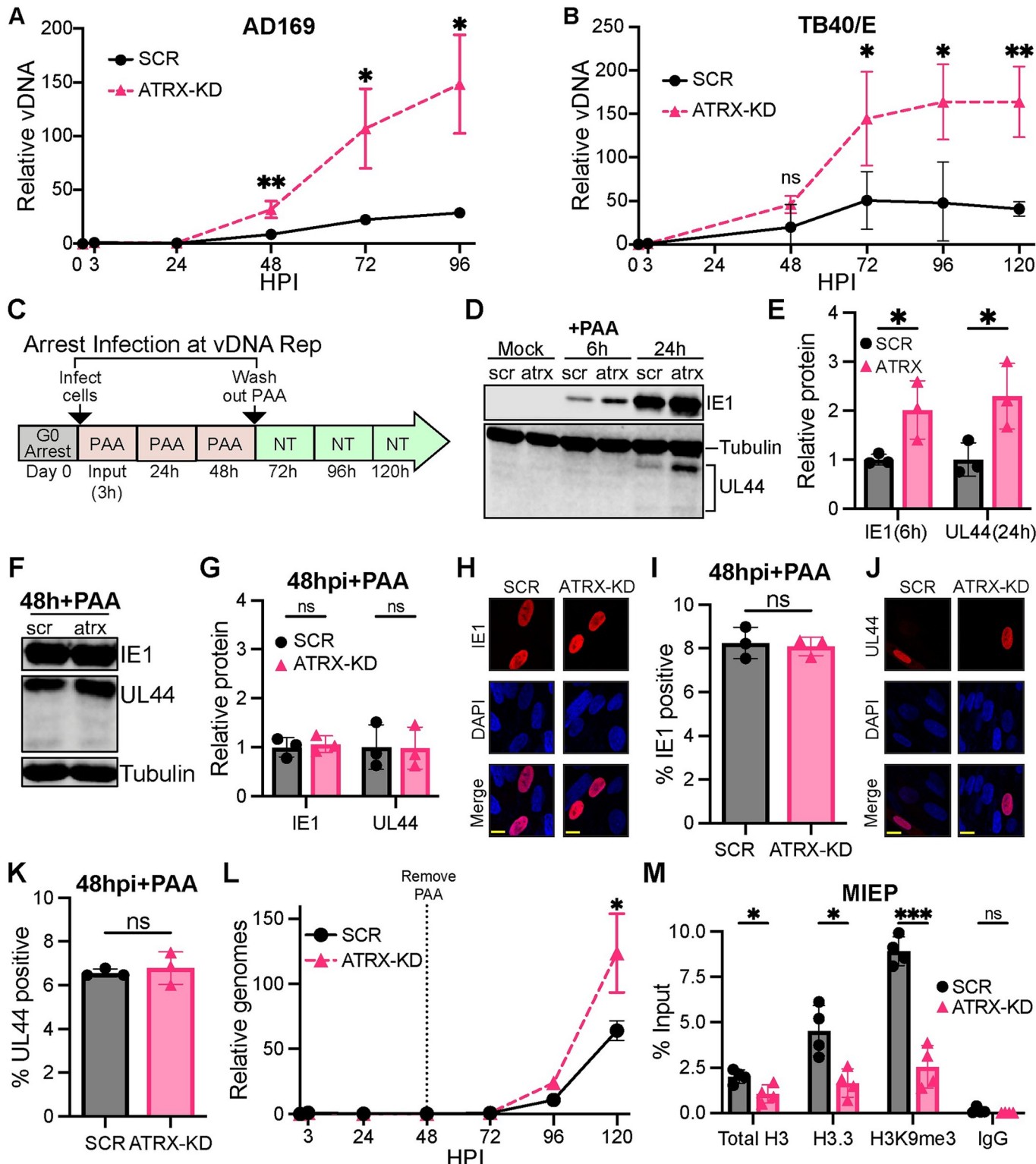

**Fig 6. Depletion of ATRX increases HCMV viral DNA replication independent of the increase in IE gene expression. A)** HFFs were infected with AD169 at an MOI of 0.1 and harvested at the indicated time points. Total DNA was analyzed by qPCR for viral genomes (MIEP) and normalized to cellular DNA (GAPDH). Plotted values are relative to respective inputs (3h). **B)** Experiments performed as in panel A with the TB40/E strain of HCMV. **C)** Summary of experimental approach for PAA experiments. HFF cells were either mock infected or infected with AD169 at an MOI of 0.1 and treated with PAA at 100ug/mL. 48 hpi after infection PAA was washed out and cells were not treated (NT) with any drug for the remaining time points. Total DNA was harvested at indicated

time points. **D and F**) Western blot in HFF SCR (scr) and ATRX-KD (atrx) for the indicated proteins in mock-infected and cells infected with AD169 for the indicated time in the presence of 100µg/mL PAA. **E**) Quantification of blots from panel D for IE1 or UL44 at 6 hpi or 24 hpi respectively. Values are normalized to tubulin and plotted relative to SCR. **G**) Quantification of blots from 48 hpi in panel F for IE1 or UL44 normalized to tubulin and plotted relative to SCR. **H-K)** HFF cells were infected with AD169 at an MOI of 0.1 in the presence of PAA, fixed at 48 hpi, and analyzed by immunofluorescence. **H and J**) Representative images of viral proteins (red) and DAPI-stained nuclei (blue) are shown. Yellow scale bars represent 10µm. **I and K**) The percentage of cells positive for the indicated HCMV protein at 48 hpi in the presence of PAA. Each replicate represents a minimum of 100 cells. **L**) Total DNA was analyzed as described in panel C by qPCR for viral genomes (MIEP) and normalized to cellular DNA (GAPDH) at the time points indicated. Plotted values are relative to respective inputs (3h). **M**) HFF cells were infected with AD169 at an MOI of 0.1 for 48 hours in the presence of 100ug/mL PAA. ChIP was performed with the indicated antibodies and analyzed by qPCR. Total H3, H3.3, H3K9me3, and IgG are plotted as percent input. All experiments were performed with a minimum of 3 biological replicates. Error bars represent standard deviation and statistically significant differences are indicated with asterisks (* = P<0.05, ** = P<0.01, *** = P<0.001; t-test).

However, after 48h in PAA, ATRX-KD and control cells showed equivalent levels of IE1 and UL44 proteins (Fig 6F and 6G). We also confirmed with IF that the same percentage of cells were IE1-positive (Fig 6H and 6I) and UL44-positive (Fig 6J and 6K) at 48hpi in the presence of PAA. Upon release from the PAA block, ATRX-KD cells still showed increased viral DNA replication compared to control cells (Fig 6L) even though they contained indistinguishable levels of IE1 and UL44 and the same percentage of cells were expressing IE1 and UL44. We conclude that ATRX impacts HCMV viral DNA replication independent of its effects on viral transcription and viral protein accumulation.

To determine if ATRX restricts HCMV viral genome replication through heterochromatinization, we used ChIP to quantitate histone occupancy and epigenetic modification at the time of DNA replication in the presence or absence of ATRX. We analyzed cells infected with HCMV for 48h in the presence of PAA to prevent DNA-replication-dependent histone deposition and to control for the increased viral DNA in ATRX-KD cells at 48 hpi (Fig 6A). Total histone H3, H3.3, and H3K9me3 occupancy was decreased in ATRX-depleted cells compared to control cells (Fig 6M). Taken together, we conclude ATRX decreases HCMV viral DNA replication by depositing and maintaining heterochromatin on the viral genome.

We reasoned that increased HCMV viral DNA accumulation in the absence of ATRX could result from an increased number of genomes replicating, and/or an increased rate of replication. To determine the number of viral genomes replicating, we quantitated the number of UL44-positive replication factories in ATRX-positive or -depleted cells because, for HSV-1 (and presumably other herpesviruses), most replication compartments emerge from a single viral genome [57–59]. Our measurements indicated that our 0.1 MOI infections delivered, on average, 4 genomes per cell (Fig 7A) a number in agreement with a previous report [60]. During infection of ATRX-positive (SCR) cells, we visualized (Fig 7B) and quantitated (Fig 7C and 7D) fewer UL44-positive replication factories than during infections of ATRX-depleted cells. More ATRX-depleted cells displayed UL44-positive foci than ATRX-positive cells (Fig 7C), and the number of UL44 foci per cell also increased in the absence of ATRX (Fig 7D). We conclude that ATRX suppresses the number of viral genomes that can initiate replication and form replication factories.

Single-molecule DNA fiber analysis [61] was then used to calculate the rate of HCMV viral DNA replication. Experiments were performed in SCR and ATRX-KD cells at 48hpi after sequential 20min pulses of IdU followed by CldU (Fig 8A). Cellular DNA synthesis is blocked in contact inhibited fibroblasts, and in the presence of PAA, we found no labeled fibers in either HCMV-infected cell type (Fig 8B), confirming the labeled fibers as HCMV viral DNA (Fig 8B and 8C). ATRX-KD cells had significant increases in the incorporation of both IdU and CldU (Fig 8C, 8D and 8E) and in the calculated rate of viral DNA replication (328 bp/min) compared to SCR cells (204 bp/min) (Fig 8F). Our calculated individual molecule DNA

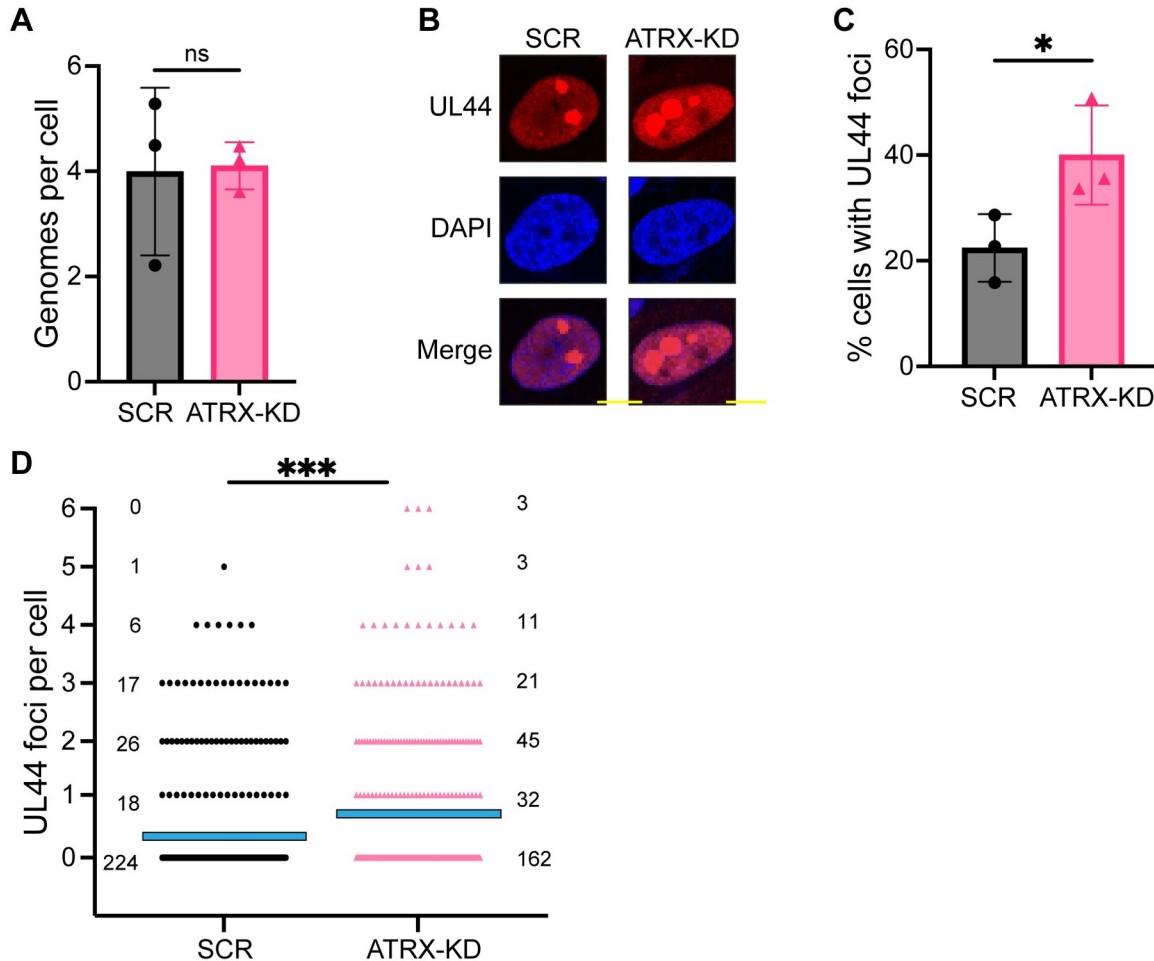

**Fig 7. ATRX knockdown increases the initiation of viral DNA replication. A)** HFF cells were infected with AD169 at an MOI of 0.1 for 3hpi. Total DNA was harvested and absolute quantification by qPCR was performed for viral genomes (IE1) per two copies of cellular DNA (GAPDH). **B)** HFF cells infected with AD169 at an MOI of 0.1 were fixed at 48 hpi and analyzed by immunofluorescence. Representative images of UL44 foci (red) and DAPI-stained nuclear DNA (blue) are shown. Yellow scale bars represent 10μm. **C)** The percentage of HCMV-infected cells with UL44 foci as imaged in panel B is shown. **D)** The number of UL44 foci per HCMV infected cells as imaged in panel B is shown. Each point represents an individual nucleus with a minimum of 50 nuclei per replicate. The mean is represented by a blue line. The number of cells is indicated next to the data points. All experiments were performed with a minimum of 3 biological replicates. Error bars represent standard deviation and statistically significant differences are indicated with asterisks (* = $P<0.05$, ** = $P<0.01$, *** = $P<0.001$; t-test (panel A and C) or Mann Whitney (panel D)).

replication rate is somewhat slower but generally agrees with a recently determined population average HCMV DNA replication rate at a higher MOI [62]. To further confirm that the only active DNA synthesis was viral we labeled nascent DNA with EdU and performed Click-chemistry combined with IF. We found that sites of EdU incorporation colocalized with UL44 viral replication compartments (Fig 8G, 8H and 8I). In PAA treated cells, UL44 staining was pan-nuclear and did not form replication factories, nor was any EdU incorporation detected in these cells (Fig 8G). We conclude ATRX suppresses the rate of HCMV viral DNA synthesis.

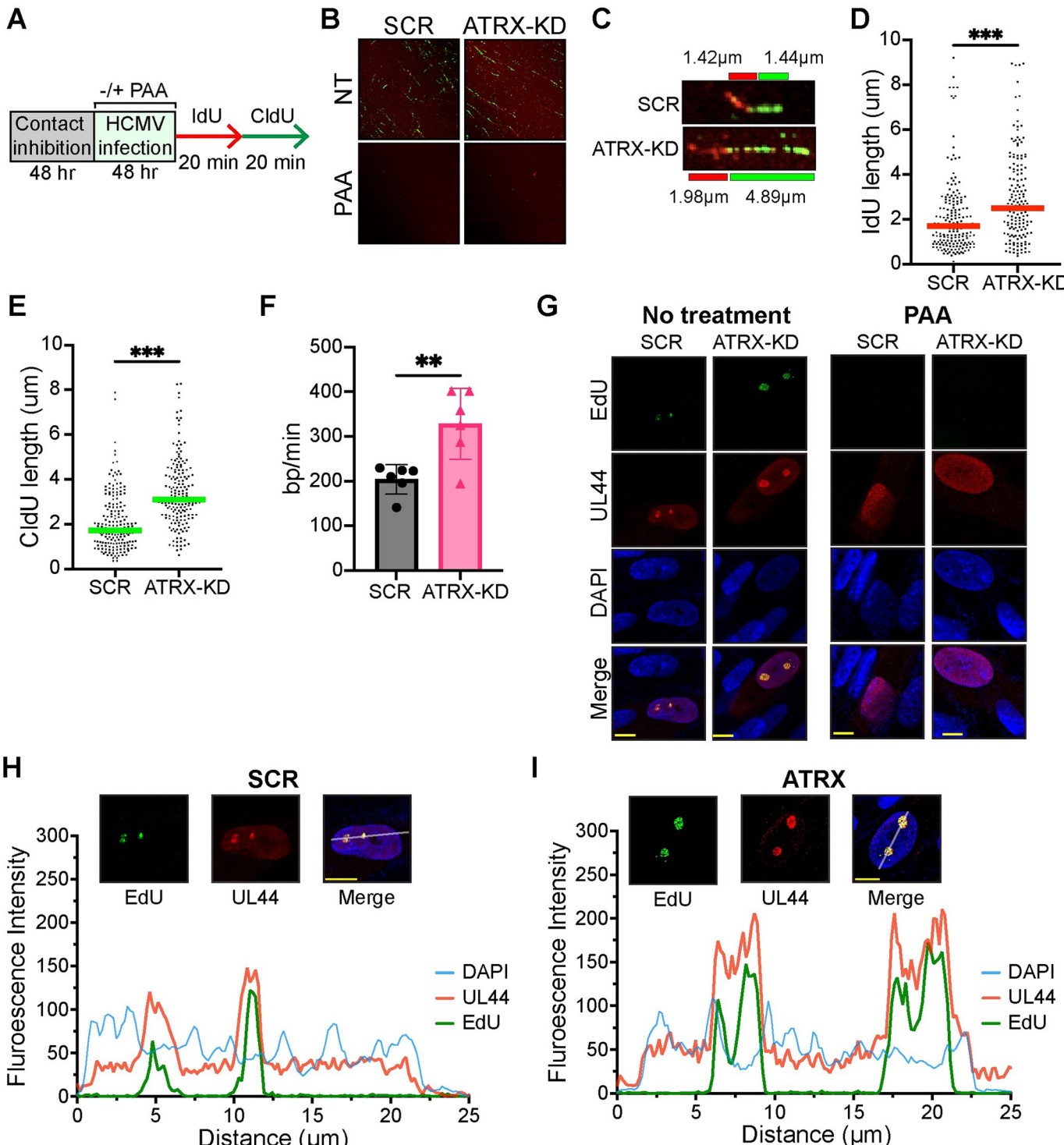

**Fig 8. Depletion of ATRX in fibroblasts increases the rate of HCMV DNA replication. A)** Schematic for single-molecule DNA fiber analysis. HFF cells were fully contact inhibited and then infected with AD169 at an MOI of 0.1 for 48hr in the absence (NT) or presence of 100μg/mL PAA. Cells were then labeled with IdU (red) and CldU (green). **B)** Representative images of DNA fibers in the indicated samples with or without PAA. **C)** Representative images of individual DNA fibers. Values represent measured fiber lengths in μm. **D)** Each point represents the length of a single IdU-labeled DNA fiber. At least 140 individual fibers were measured for each condition. Similar results were obtained for three independent biological replicates of HCMV infection. **E)** CldU fiber analysis as in panel D. **F)** Mean fiber lengths determined as in panels D and E are plotted as DNA base pairs per minute (bp/min) for each biological replicate. **G-I)** HFF cells infected with AD169 at an MOI of 0.1 and either untreated or PAA treated. Cells were then labeled with EdU at 48hpi, fixed and EdU was labeled with Alexa Fluor 488 by a click reaction. Cells were then analyzed by immunofluorescence. **G)** Representative images of UL44 foci (red), EdU-labeled DNA (green), and DAPI-stained nuclear

DNA (blue) are shown. Yellow scale bars represent 10μm. **H and I)** Zoomed-in representative images of nuclei from panel G of indicated cell type. Yellow bars represent 10μm and white bars represent a 25μm path of fluorescent intensity analysis across a lateral section. All experiments were performed with a minimum of 3 biological replicates. Error bars represent standard deviation and statistically significant differences are indicated with asterisks (* = P<0.05, ** = P<0.01, *** = P<0.001; Mann Whitney (panel D and E) or t-test (panel F)).

## ATRX prevents the accumulation of excess intracellular unpackaged viral DNA

Finally, we asked whether the ~5-fold increase in viral DNA accumulation was converted to a similar increase in infectious progeny virion formation. Interestingly, ATRX-KD cells show only a modest, ~2-fold increase in infectious progeny compared to control cells for AD169 infected cells and no increase in TB40/E infected cells (Fig 9A). Lower-than-expected titers may result if the increased rate of viral DNA replication made it more error prone, creating defective genomes. If this were true, virions produced from ATRX KD cells may have lower PFU/genome ratios than those produced in SCR cells. Therefore, we quantitated viral DNA in extracellular virions as well as the infectivity of those virions and compared the two values. We found indistinguishable infectivity of packaged viral genomes in extracellular virions released from ATRX-KD or SCR cells for both AD169 and TB40/E (Fig 9B). We conclude the disparity between infectious progeny release and viral DNA accumulation in ATRX-KD cells is not the result of increased amounts of defective genome containing particles.

We conducted similar experiments quantitating viral DNA in cellular lysates (as opposed to extracellular virions) as well as the associated infectivity of virions isolated from those lysates. Here we found that each genome in ATRX-KD cells produced fewer PFUs than in SCR cells (Fig 9C). Because the genomes in virions released from ATRX-KD cells are not defective (Fig 9B), we conclude that ATRX-KD cells have a defect in a viral process sometime between viral DNA replication and infectious progeny release, events collectively termed assembly and egress. Because ATRX is a nuclear protein and directly impacts the viral genome (Figs 4 and 5), we focused on packaging, the major nuclear assembly and egress event that directly involves the viral genome.

HCMV DNA replication results in the formation of multi genome length concatemers [63] that are converted to unit-length genomes during packaging into capsids by site-specific cleavage at a distinct locus defined by the *pac* motifs [64–66]. Cleavage is catalyzed by the viral terminase complex that associates with the capsid portal and drives genome packaging [66,67]. We monitored HCMV genome cleavage as a direct assay for the impact of ATRX knockdown on the efficiency of packaging. In unpackaged, concatemeric (uncleaved) viral DNA, the pac sequence resides in an ~8kb KpnI restriction fragment. In virion-packaged unit length linear genomes, that fragment is cleaved so that the pac sequence now resides in an ~4kb fragment (Fig 9D). These two differently sized fragments can be easily distinguished and quantitated by Southern blotting (Fig 9E) with an appropriately specific probe using a well-established assay [68].

Total HCMV DNA (cleaved + uncleaved) was 1.8-fold higher in ATRX-KD cells compared to SCR cells (Fig 9F), in agreement with our previous assays (Fig 6A). As expected, cleaved viral DNA levels were lower (Fig 9F) because not all viral DNA in a cell is packaged into capsids (Fig 9D). Importantly, cleaved viral DNA was only 1.2-fold higher in ATRX-KD cells compared to SCR cells (Fig 9F), indicating a smaller fraction of viral DNA in ATRX cells was cleaved (packaged) compared to SCR cells. We used these data to calculate the ratio of cleaved genomes compared to total viral DNA (cleaved + uncleaved) and present the data as the percent of total viral DNA within a cell that is packaged (packaging efficiency). We found a

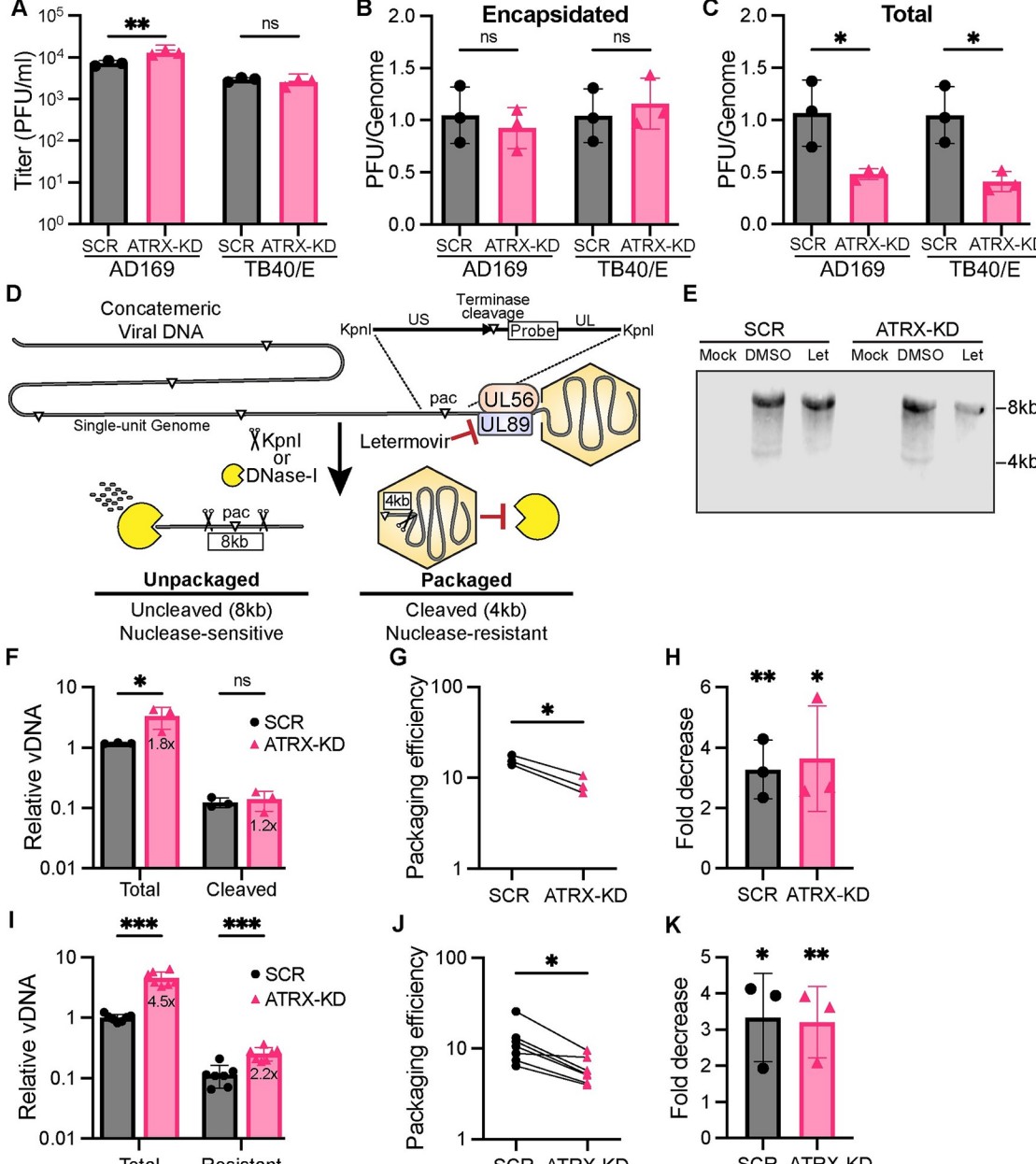

**Fig 9. Depletion of ATRX in fibroblasts decreases the efficiency of viral genome packaging. A)** HFF cells were infected with the indicated strain of HCMV at an MOI of 0.1 and viral progeny collected at 96 hpi were quantified by plaque assay. **B)** Encapsidated viral DNA from cell-free viral progeny collected as in panel A was harvested by DNAzol extraction and absolute quantitation by qPCR (IE1) was performed. The ratio of PFU per encapsidated viral genome was analyzed and reported relative SCR. **C)** HFFs were infected as in panel A. At 96 hpi total DNA was harvested and analyzed by qPCR for viral genomes (MIEP) and normalized to cellular DNA (GAPDH). The ratio of PFU per total viral DNA was analyzed and reported relative SCR. **D)** Schematic model of HCMV DNA packaging assays. Concatemeric viral DNA genomes are susceptible to DNase-I digestion, whereas packaged DNA is protected. The HCMV terminase complex (UL56/UL89) cleaves concatemeric HCMV genomes into single units during capsid packaging, which is inhibited by Letermovir treatment. The pac sequence is recognized by the terminase complex which cleaves the genome at the terminus. A zoomed-in view of the terminus is depicted with surrounding KpnI restriction sites. Following KpnI restriction enzyme digestion, concatemeric viral DNA yields an ~8.4-kb fragment, whereas cleaved/packaged DNA results in a shortened ~4-kb fragment. Both fragments are recognized by the digoxigenin-labeled probe. **E)** Representative image of Southern blot analysis of HFF cells infected with AD169 at an MOI of 0.1 for 96 hpi and either mock-treated (DMSO) or treated with Letermovir (Let). Blotting was carried out using the terminal DNA probe depicted in panel D. **F)** Blots from panel E were quantified for total (8.4kb + 4kb) and cleaved (4kb). Plotted values are relative to SCR-Total. **G)** Data from panel F were used to calculate percent packaged by the ratio of the signal intensity of the cleaved (4kb) band over the total (8kb+4kb). **H)** Fold decrease is

calculated as the negative fold change of percent packaged DNA between DMSO and Letermovir treatment. SCR and ATRX were statistically compared to their respective DMSO control. **I)** HFFs were infected with AD169 at an MOI of 0.1 and harvested at 96hpi. Lysates were either mock-treated or treated with DNase-I. DNA was then harvested and analyzed by qPCR for viral genomes (MIEP) and normalized to mock-treated cellular DNA (GAPDH). Plotted values are relative to SCR-Total. **J)** Data from panel I for the percent packaged is calculated by the ratio of resistant DNA to total. **K)** Fold decrease is calculated as the negative fold change of percent packaged DNA between DMSO and Letermovir treatment. SCR and ATRX were statistically compared to their respective DMSO control. All experiments were performed with a minimum of 3 biological replicates. Error bars represent standard deviation and statistically significant differences are indicated with asterisks (* = P<0.05, ** = P<0.01, *** = P<0.001; t-test).

significantly lower fraction of viral DNA was packaged in ATRX KD cells compared to SCR cells (Fig 9G). Letermovir is an FDA-approved drug that inhibits the HCMV terminase protein complex, impairing viral genome cleavage and resulting in fewer (genome containing) C capsids [68]. We found Letermovir significantly and equally reduced the signal for the 4kb cleaved viral genome fragment in our Southern blot assay in both SCR and ATRX KD cells (Fig 9E and 9H), confirming its identity as a marker for cleavage (and packaging) and validating the design of this assay.

Next, we analyzed the fraction of viral DNA in SCR and ATRX-KD cells that was nuclease resistant, a property the viral genome acquires after packaging [69–71]. Lysates were prepared from HCMV infected cells and either mock-treated or treated with DNase-I before HCMV genomes were quantitated by qPCR. Under these conditions, unpackaged viral genomes would be nuclease-sensitive while packaged genomes would be protected from nuclease digestion by the capsid. Total HCMV DNA in the mock treated samples was 4.5-fold higher in ATRX-KD cells compared to SCR cells (Fig 9I), in agreement with our previous assays (Fig 6A). As expected, DNA levels were lower (but not zero) after DNase-I treatment because not all viral DNA in a cell is packaged into capsids (Fig 9D). Importantly, DNase-I resistant DNA was only 2.2-fold higher in ATRX-KD cells compared to SCR cells (Fig 9I), indicating a smaller fraction of viral DNA in ATRX cells was nuclease resistant (packaged) compared to SCR cells. We used these data to calculate the ratio of nuclease resistant viral genomes compared to total viral DNA and present the data as the percent of total viral DNA within a cell that is packaged (packaging efficiency). We found that HCMV genomes were more efficiently packaged in SCR cells compared to ATRX-KD cells (Fig 9J). As with viral genome cleavage (Fig 9H), Letermovir equally and significantly decreased the amount of nuclease-resistant viral DNA in both SCR and ATRX KD cells (Fig 9K) confirming the assay accurately quantitates packaging. We conclude that HCMV genome packaging occurs less efficiently in ATRX-KD cells and that ATRX restriction of viral DNA replication acts in a pro-viral manner to suppress the overproduction of viral genomes that would exceed packaging capabilities. In total, we conclude that ATRX acts in an antiviral mode at the start of productive HCMV infections to repress viral transcription but in a pro-viral manner at later times to better equate viral genome production with capsid packaging limits (Fig 10).

## Discussion

HCMV is a betaherpesvirus with a broad cellular tropism and disseminates throughout the body leading to multiorgan disease [72,73]. HCMV is the leading cause of virus-induced birth

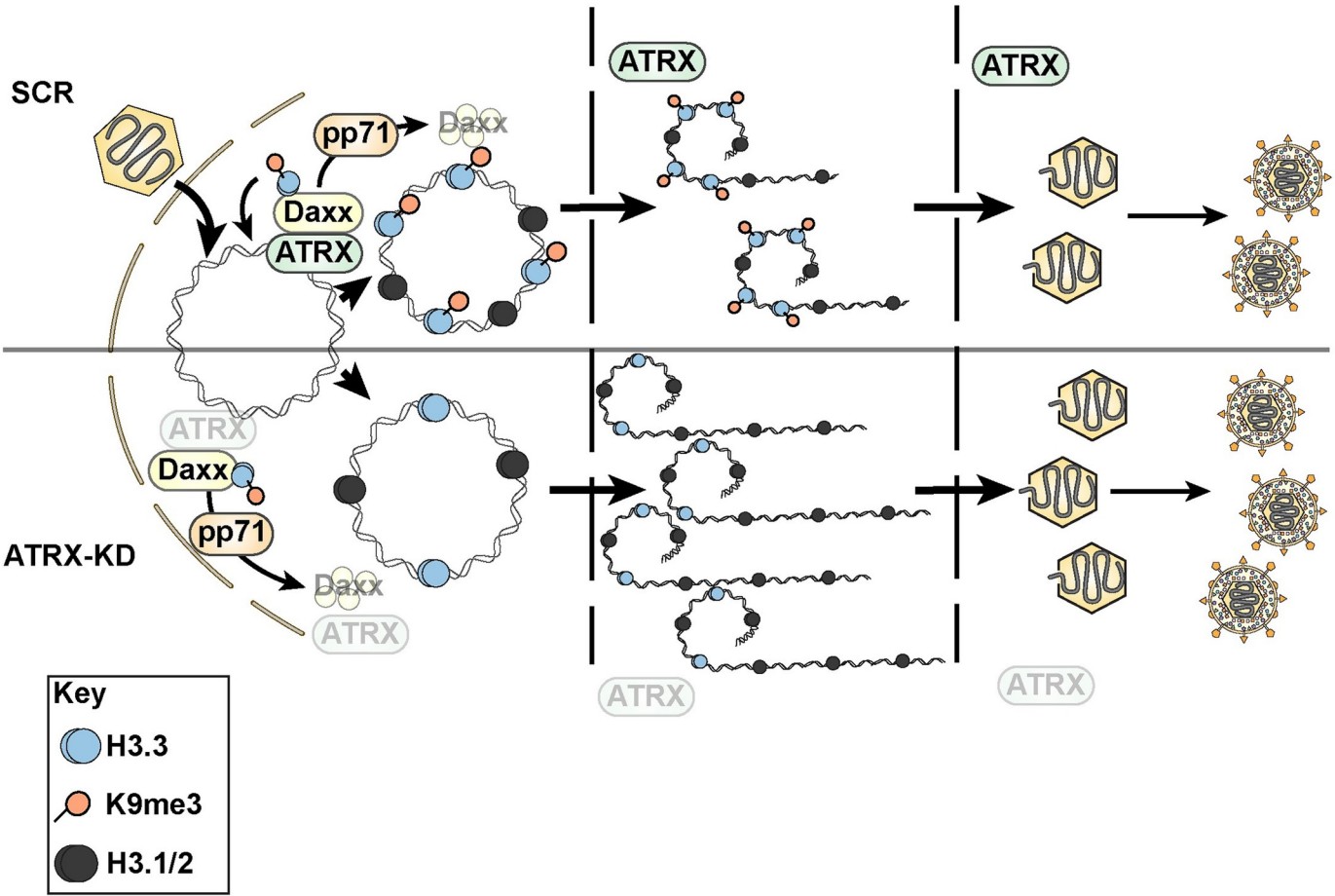

**Fig 10. Model for anti- and pro-viral effects of ATRX on HCMV lytic replication. Top**. ATRX heterochromatinizes viral genomes and restricts viral transcription at immediate early (IE) times acting in an anti-viral manner unless inactivated by the pp71-mediated degradation of Daxx. However, at early times HCMV capitalizes on ATRX heterochromatinization to prevent the over-replication of viral genomes. Here, ATRX acts in a pro-viral manner to allow for efficient viral genome packaging into capsids at Late times. **Bottom**. In ATRX-depleted cells, viral IE transcription and DNA replication are both enhanced, but viral genomes are packaged less efficiently.

defects [74,75] is a major contributor to complications in organ transplants (e.g., organ rejection, pneumonia, decreased survival) [76,77], and is associated with glioblastoma multiforme (GBM) tumors [78–81]. Productive (lytic) infection causes disease, but the virus can also establish a latent reservoir that permits both lifelong infection and sporadic reactivation to lytic replication [82]. There is no vaccine for HCMV, and while the currently available antiviral treatments inhibit productive replication, they select for resistant variants and fail to target latently infected cells [83,84]. As such, novel antiviral interventions are needed.

One promising new therapeutic avenue targets viral epigenetics to either induce latent cells to reactivate into the lytic cycle for which there are effective antivirals (shock and kill) or to drive lytic infections into a latency-like state and prevent them from reactivating (block and lock) [85,86]. Epigenetics [87,88] refers to the architecture of protein-DNA (chromatin) complexes that impact transcriptional outputs and can thereby determine infection outcome (lytic or latent). Indeed, the chromatin structure of the HCMV viral double-stranded DNA genome is well established to control viral transcription during both lytic infection and latency [89–

91]. A better understanding of the processes that create, maintain, and remodel viral chromatin structure is needed to fully implement epigenetic therapies.

Here we show that ATRX restricts HCMV IE transcription by decreasing the accessibility of the MIEP through the deposition of histones marked with transcriptionally repressive post-translational modifications. Therefore, both ATRX and Daxx, as well as writers, readers, and erasers of the H3K9me3 histone mark are candidates for targeting with antiviral epigenetic therapies. Perhaps more importantly, our work reveals a later, positive role for ATRX during HCMV infection. In ATRX-depleted cells, viral DNA accumulates rendering packaging inefficient, which may result in a lack of viral fitness *in vivo*. Furthermore, excess unpackaged viral DNA could be immunogenic if released from dying or dead cells. Thus, HCMV may capitalize on the functions of ATRX to limit its viral DNA replication to a level commensurate with the packaging capabilities of the virus. ATRX may have additional effects on steps of virion assembly and egress subsequent to packaging, perhaps through its ability to impact the expression of cellular innate immunity genes [45].

We achieved similar results for the impact of ATRX on two different strains of HCMV, the fibroblast-adapted ("laboratory") strain AD169 and the low passage ("clinical") strain TB40/E. The only significant difference was the inability of ATRX-KD to impact total histone H3 association with the viral genome for AD169 (Fig 5B), whereas it decreased total H3 association with TB40/E (Fig 5C). Importantly, ATRK-KD decreased the association of histone H3.3, the histone it helps deposit, with both AD169 and TB40/E genomes. Total H3 includes histone H3.1 and H3.2. We have shown that H3.1 is associated with HCMV genomes [21], but do not know the histone chaperone that deposits it, nor how that chaperone may be targeted to the viral genome. Perhaps whatever feature prompts H3.1 deposition onto HCMV genomes is a more prominent feature of AD169 compared to TB40/E. Our ATAC-seq analysis suggested that ATRX is recruited to GC-rich regions of the HCMV genome. However, the MIEP is not particularly GC-rich, but does contain an ATRX-recruiting RUNX1 binding site [56]. Provocatively, only the RUNX1 binding sites near the terminal repeats displayed increased accessibility in the absence of ATRX, implicating the ends of the viral genome as participating in ATRX-mediated restriction. More work is needed to define how ATRX broadly impacts HCMV genome accessibility while also homing in on certain loci, like the MIEP and terminal repeats.

In HSV-1 lytically infected cells, ATRX levels decrease and remain low well past the time when viral genomes are replicated [3]. Interestingly, replicated HSV-1 genomes are not histone-associated [16], and the reported rate of HSV-1 viral DNA synthesis (1322–1919 bp/min) [92] is substantially higher than what we (Fig 8F) and others [62] measured for HCMV. Accumulating higher levels of viral DNA may not be an issue for HSV-1 as its genome is much smaller than HCMV, is packaged less densely into capsids than is the HCMV genome [93], and the virus grows to much higher titers *in vitro*. Furthermore, the much faster replication cycle of HSV-1 may allow it to outpace any immune activation potentially elicited by unpackaged viral genomes. Thus, an ATRX-mediated decrease in viral DNA accumulation may not be pro-viral for HSV-1 (and in fact may be antiviral), perhaps explaining why HSV-1 targets the ATRX protein for ICP0-dependent degradation and the ATRX encoding mRNA for viral miRNA-dependent translational silencing.

ATRX has multiple ways through which it could slow down HCMV DNA replication. One is through histone deposition of H3.3 in cooperation with Daxx. While degraded by pp71 at the start of lytic infection, Daxx reaccumulates by the time viral DNA replication initiates, and replicated HCMV genomes, unlike HSV-1, are histone-associated [15]. Thus, Daxx may be allowed to re-accumulate during HCMV infection to permit the Daxx-ATRX complex to slow viral DNA replication, allowing for efficient viral genome packaging. Another mechanism

through which ATRX could slow down viral DNA replication is if it participated in the writing of H3K27me3 marks on histones associated with the viral genome as it does for the X chromosome [43,44]. The abilities of ATRX to impair histone macroH2 deposition or to resolve G4 quadruplex structures seems unlikely to be responsible for its negative impact on HCMV viral DNA replication. In fact, ATRX resolution of G4 quadruplexes might even be expected to accelerate replication of the template, although perhaps the act of binding and resolution is sufficient to slow viral DNA replication. Lastly, but perhaps most intriguingly, the ability of ATRX to increase the expression of cellular genes downstream of innate immune activation independent of Daxx during HCMV infection [45] may impact viral DNA replication. Indeed, of all the biological processes mediated by the genes directly regulated by ATRX, the single highest confidence GO term was DNA replication [45]. More work is needed to determine the mechanism through which ATRX suppresses HCMV viral DNA replication.

In summary, our study underscores ATRX's critical role in modulating HCMV lytic infections. ATRX initiates heterochromatinization of viral genomes, effectively restricting viral IE transcription and decelerating viral DNA replication. Our novel finding that ATRX positively impacts HCMV genome packaging efficiency may explain the different mechanisms used by HCMV and HSV-1 to target the Daxx-ATRX complex. Finally, the anti-viral functions of ATRX at the start of lytic HCMV infections and its potential pro-viral functions at later time points present an exciting new perspective on the role of PML-NB proteins during herpesvirus infections and illuminate promising intervention points for antiviral strategies.

## Methods

### Cells and infections

TERT-immortalized human foreskin fibroblasts (HFFs) were maintained in Dulbecco's modified Eagle medium (DMEM; Gibco). Letermovir (Let) (500nM; SML3891; Sigma) was added to HFF cells after viral inoculum was removed. Medium was supplemented with 10% fetal bovine serum (FBS; GeminiBio), 100 U/mL penicillin, 0.1 mg/mL streptomycin, and 2 mM l-glutamine (PSG; Sigma). All viruses used were derived from BACs electroporated into deidentified primary Normal Human Dermal Fibroblasts (NHDFs; Clonetics) with a pp71 expression construct. Virus was concentrated by ultracentrifugation through a 20% sorbitol cushion. Titers were calculated by plaque assay on naive NHDFs. For HCMV infections, cells were grown to superconfluency and given fresh media 24hrs before infection. Cells were then incubated with virus in a small volume for 1 hr at 4 ˚C, then moved to 37˚C for 15 min to allow for synchronized viral entry. Viral inoculum was removed, and medium was replaced to normal volume.

### Inhibitors and antibodies

Phosphonoacetic acid (PAA) (100 μg/mL; Sigma) was added to HFF cells after viral inoculum was removed. The following commercial antibodies were used for Western blotting: anti-ATRX (ab97508; Abcam), anti-DAXX (D7810; Sigma), anti-UL44 (CA006-100; Virusys) and anti-tubulin (DM1A; Sigma). For ChIP: anti-Histone H3 (Total H3) (ab1791; Abcam), anti-Histone H3.3 (ab176840; Abcam), and anti-Histone H3K9me3 (ab176916; Abcam). For immunofluorescence, IE1 (clone 1B12, [94]), anti-UL44 (CA006-100; Virusys) Monoclonal antibodies for Western blotting against IE1 (clone 1B12, [94]) and pp150 (CMV127, [95]) have been previously described. For DNA fiber assay: Rat anti-BrdU (ab6326; Abcam), Mouse anti-BrdU (347580; BD Biosciences), anti-mouse IgG1 Alexa Fluor 568 (A11004; Invitrogen), and anti-rat Alexa Fluor 488 (A11006; Invitrogen).

### Knockdown cell lines

ATRX knockdown (ATRX-KD) fibroblast cells were generated by lentiviral transduction of a shRNA expression construct targeting ATRX. Briefly, recombinant lentivirus was produced by cotransfection of 293T cells with pLKO.1-shATRX (Thermo TRCN0000013590), pMD-VSV-G, and pPAX2. HFFs were transduced with recombinant lentivirus and selected for successful transduction with 5 ug/mL puromycin and maintained in 1 ug/mL puromycin. To generate a shRNA control cell line, a pLKO.1 vector expressing a scramble shRNA sequence was used (SCR) [96].

### Knockout cell lines

Synthetic single guide RNA (sgRNA) targeting ATRX (ucuacgcaaccuuggucgaa) was ordered from Synthego [30]. sgRNAs were cotransfected with Cas9 mRNA (L-7606-100; TriLink) into HFF cells. DNA was collected from the bulk population, and PCR was performed across the sgRNA cut site and sent for Sanger sequencing. Results were analyzed by Tracking of Indels by Decomposition (TIDE) [97]. After 72 hrs of recovery, single cells were isolated by Flow-Assisted Cell Sorting (FACS) into 96 well plates. Cell lines were screened by PCR across the sgRNA cut site and sequenced by Sanger sequencing.

### Western Blotting

Cells were lysed in 1% SDS containing 2% BME. Lysates were run on SDS-PAGE gels, transferred on Optitran membranes (GE Healthcare), and analyzed by Western blotting.

### DNA and mRNA analysis

Total genomic DNA was harvested from cells using the Mini Genomic DNA kit (blood & cultured cells) (IB47200; IBI). Total RNA was harvested using Mini Total RNA kit (blood & cultured cells) (IB47320: IBI). Equal amounts of total RNA were used to generate cDNA with the Maxima H Minus cDNA Synthesis Master Mix kit with dsDNase kit (M1682; Thermo). Encapsidated viral DNA was harvested from infected cell supernatant treated with micrococcal nuclease (M0247S; NEB) and then extracted with DNAzol (10503027; Thermo) as previously described [98]. DNA and cDNA were analyzed by quantitative PCR using iTaq Universal SYBR green Supermix (1725124; BioRad). Normalization was performed using the ΔCt method [99]. Absolute quantitation was performed by generating a standard curve with known copy number plasmids of IE1 and GAPDH and then comparing samples to the standard curve to extrapolate a copy number value. Primers used for qPCR are listed in Table 2.

### FAIRE Assay

Formaldehyde Assisted Isolation of Regulatory Elements was performed as previously described [100]. In brief, infected cells were fixed in 1% formaldehyde for 5 min at room temperature and quenched by adding glycine to 125 mM. Nuclei were then isolated, and DNA was sheared by sonication. Accessible DNA was isolated by phenol/chloroform extraction and precipitated by adding 2 volumes of 95% ethanol and incubating at -20 ˚C overnight. DNA was isolated using Gel Extraction & PCR Cleanup Kit (IB47010; IBI) and was analyzed by quantitative PCR using iTaq Universal SYBR green Supermix (1725124; BioRad). Normalization to a reverse-crosslinked input was performed using the ΔCt method [99].

**Table 2. List of qPCR primers used.**

| Target | Forward Primer (5' to 3') | Reverse Primer (5' to 3') |
|---|---|---|
| IEex3 | CGACGTTCCTGCAGACTATG | TCCTCGGTCACTTGTTCAAA |
| UL44 | AGCCGCACTTTTGCTTCT TG | TCGCAACTCCGGCAATTA CT |
| pp150 | AACCTCTTCCGCTTCTTC | GGACACGACATCATCCTC |
| GAPDH | GAGCCAAAAGGGTCATC | GTGGTCATGAGTCCTTC |
| MIEP | CTTATGGGACTTTCCTACTTG | CGATCTGACGGTTCACTAA |
| Actin-B Promoter (ACTB) | AAAGGCAACTTTCGGAACGG | TTCCTCAATCTCGCTCTCGC |
| HeterochromatinChr.12 (Chr.12) | ATGGTTGCCACTGGGGATCT | TGCCAAAGCCTAGGGGAAGA |
| OriLyt | GACGGCTTCCGGGTCT | GCCGGACCCTCGAGAG |
| UL99/pp28 | CGTCTCTACCGTGCTAGACC | CATCTTTCAGGGGCTCACCG |

## ATAC-seq

ATAC libraries were generated using Zymo-Seq ATAC Library Kit (D5458; Zymo). Libraries were sequenced by the Biotechnology Center at the University of Wisconsin-Madison using 2x150bp on an Illumina NovaSeq. For analysis, the human genome (hg38) was combined with a modified AD169 genome (GenBank: FJ527563.1), where the internal repeats were deleted. Reads were aligned using the PEPATAC pipeline [101]. Briefly, read quality was assessed with FastQC, and reads were trimmed to remove adapters with CutAdapt [102]. Trimmed reads were aligned with Bowtie2 [103] first to the mitochondrial genome, and unaligned reads were rerun with Bowtie2 (—very-sensitive -X 2000) to the combined hg38 and AD169 genome. Deduplicated and high-quality aligned reads were then used to call peaks using MACS2 (—shift -75—extsize 150—nomodel—call-summits—nolambda -p 0.01). Normalized coverage maps were generated using DeepTools bamCoverage [104] and visualized with Integrated Genome Viewer (https://igv.org/). Differential coverage was performed using UCSF bigWigA-verageOverBed [53] and the mean coverage across each HCMV feature was compared between SCR and ATRX-KD samples. Motif analysis was performed with the MEME suite [54].

## ChIP Assays

Chromatin immunoprecipitation assays were performed as previously described [90]. Briefly, infected cells were fixed in 1% formaldehyde for 8 min at room temperature and quenched by adding glycine to 125 mM. Nuclei were then isolated, and DNA was sheared by sonication. Sonicated DNA was incubated overnight with 2 μg of antibody rotating at 4°C, then immuno-precipitated with protein A/G beads (88803; Thermo) for 1 hr rotating at 4 °C. Beads were washed, reverse crosslinked, and DNA was eluted from beads. DNA was isolated using Gel Extraction & PCR Cleanup Kit (IB47010; IBI) and was analyzed by quantitative PCR using iTaq Universal SYBR green Supermix (1725124, BioRad). Normalization to input was performed using the ΔCt method [99].

## Immunofluorescence

Cells were seeded on coverslips and infected for 48 hrs. Cells were then fixed in 1% formaldehyde for 30 min at room temperature. Cells were permeabilized with PBST (PBS with 0.1% Triton X-100 and 0.05% Tween 20), blocked for 30 min in PBST plus 0.5% goat serum and 0.5% bovine serum albumin. Cells were then incubated with primary antibody and subsequently anti-mouse Alex Fluor 568 (1:1000) for 1 hr each at room temperature in blocking

buffer. Nuclei were then stained with DAPI (1:10000) for 5 min, and coverslips were mounted with Fluoromount-G (00-4958-02; Invitrogen). Images were taken with a Leica Stellaris confocal microscope using a 63X oil immersion objective lens at 1X digital zoom and 1024X1024 resolution.

## DNA fiber assay

Single molecule DNA fiber assays were performed as previously described [61]. Briefly, infected cells were pulsed in 20mM IdU for 20 min, washed, and subsequently pulsed with 50mM CldU for 20 min. Cells were harvested by trypsinization, pelleted, and resuspended in media. The cells were then pipetted onto positively charged slides and DNA Lysis Buffer was added to lyse the cells for 5 min. Slides were tilted at sixty degrees to spread genomic DNA and dried for 15 min. DNA was fixed in 3:1 methanol:acetic acid solution and then denatured in 2.5 M Hydrochloric acid for 1 hour. Slides were blocked in 3% BSA in PBS for 30 min. The denatured nascent DNA was stained with primary antibodies [Abcam rat anti-BrdU (1:1000) and BD Biosciences mouse anti-BrdU (1:500)] for 30 min at room temperature. Slides were washed in PBST buffer 3 times to wash the unbound antibodies. Slides were then stained with fluorophore-conjugated secondary antibodies [1:1000 dilution of anti-rat Alexa Fluor 488 and anti-mouse IgG1 Alexa Fluor 568] for 30 min followed by washes with PBST. Cover slips were affixed to slides using ProLong Gold Antifade Mountant (P10144; Invitrogen). Fibers were imaged with a Leica Stellaris confocal microscope using a 63X oil immersion objective lens at 1X digital zoom and 1024X1024 resolution. The lengths of the DNA fibers were measured using ImageJ software. IdU/CldU incorporation rates were measured in base pairs per minute using a conversion factor of 2.59 kb/μm of DNA fiber.

## Nascent DNA Labeling with EdU

Cells were seeded on coverslips and infected for 48 hrs. Cells were then labeled with 10μM EdU in media for 2 hrs at 37°C. EdU was visualized with the Click-iT EdU Cell Proliferation Kit for Imaging, Alexa Fluor™ 488 dye (C10337; Invitrogen). Cells were then incubated with anti-UL44 antibody and subsequently anti-mouse Alex Fluor 568 (1:1000) for 1 hr each at room temperature in blocking buffer. Nuclei were then stained with DAPI (1:10000) for 5 min, and coverslips were mounted with Fluoromount-G (00-4958-02; Invitrogen). Images were taken with a Leica Stellaris confocal microscope using a 63X oil immersion objective lens at 1X digital zoom and 1024X1024 resolution. Fluorescence intensity was measured using plot profile with Image J software.

## DNase-I resistance assay

Cells were harvested and lysed by three freeze/thaw cycles. Cells were then split into two aliquots. One aliquot was digested with DNase-I (NEB M0303S) for 2 hours at 37°C, and the other was left untreated. DNA was then harvested from both aliquots using the Mini Genomic DNA kit (blood & cultured cells) (IB47200; IBI). DNA was analyzed by quantitative PCR using iTaq Universal SYBR green Supermix (1725124; BioRad). Normalization was performed using the ΔCt method [99].

## Southern blotting

Southern blotting was performed as previously described [105]. Briefly, cells were infected and total genomic DNA was harvested from cells using the Mini Genomic DNA kit (blood & cultured cells) (IB47200; IBI). 10μg of DNA was digested with KpnI (NEB R3142) for 2hrs at

37˚C. Digested DNA was run on an agarose gel and then transferred to a positively charged nylon membrane (AM10100; Invitrogen). Digoxigenin (DIG) labeled DNA probes were generated using the PCR DIG Probe Synthesis Kit (11636090910; Roche) targeting terminase-cleaved and uncleaved (AD169 genome positions 703 to 1524) [68]. Hybridization was then performed in DIG Easy Hyb Buffer (11603558001; Roche). Membranes were incubated with anti-Digoxigenin-AP, Fab fragments (11093274910; Roche) and developed using CDP-Star Substrate (1:3000) (T2304; Thermo). Blots were imaged and analyzed using an Odyssey Fc imager with Image Studio Lite v.5.2.5 software (LI-COR).

## Acknowledgments

We thank all members of the Kalejta Laboratory for helpful discussions, especially Emily Albright for her tireless efforts in support of the lab.

## Author Contributions

**Conceptualization:** Ryan M. Walter, Robert F. Kalejta.

**Data curation:** Ryan M. Walter, Kinjal Majumder, Robert F. Kalejta.

**Formal analysis:** Ryan M. Walter, Robert F. Kalejta.

**Investigation:** Ryan M. Walter.

**Methodology:** Ryan M. Walter, Robert F. Kalejta.

**Visualization:** Ryan M. Walter.

**Writing – original draft:** Ryan M. Walter, Robert F. Kalejta.

**Writing – review & editing:** Ryan M. Walter, Robert F. Kalejta.

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
