## [Decision Letter · Decision Letter 0]

6 May 2024

Dear Dr. Kalejta,

Thank you very much for submitting your manuscript "1 ATRX restricts Human Cytomegalovirus (HCMV) viral DNA replication through heterochromatinization and minimizes unpackaged viral genomes." for consideration at PLOS Pathogens. As with all papers reviewed by the journal, your manuscript was reviewed by members of the editorial board and by several independent reviewers. In light of the reviews (below this email), we would like to invite the resubmission of a significantly-revised version that takes into account the reviewers' comments.

The reviewers appreciate the identification of multiple roles for ATRX in HCMV replication. However, major concerns were noted regarding interpretations that ATRX impacts viral DNA accessibility, viral DNA replication and the packaging of viral genomes. We encourage the authors to carefully consider and perform additional experiments that are needed define the functional consequences of ATRX loss more rigorously and support the major conclusions. Text revisions to clarify methods and results are also detailed.

We cannot make any decision about publication until we have seen the revised manuscript and your response to the reviewers' comments. Your revised manuscript is also likely to be sent to reviewers for further evaluation.

Sincerely,

Laurie T Krug, PhD

Academic Editor

PLOS Pathogens

Patrick Hearing

Section Editor

PLOS Pathogens

Michael Malim

Editor-in-Chief

PLOS Pathogens

orcid.org/0000-0002-7699-2064

The reviewers appreciate the identification of multiple roles for ATRX in HCMV replication. However, major concerns were noted regarding interpretations that ATRX impacts viral DNA accessibility, viral DNA replication and the packaging of viral genomes. We encourage the authors to carefully consider and perform additional experiments that are needed define the functional consequences of ATRX loss more rigorously and support the major conclusions. Text revisions to clarify methods and results are also detailed.

Reviewer's Responses to Questions

**Part I - Summary**

Reviewer #1: The manuscript “ATRX restricts Human Cytomegalovirus (HCMV) viral DNA replication through heterochromatinization and minimizes unpackaged viral genomes” by Walter et al. focuses on the role of the cellular protein Alpha Thalassemia/Mental Retardation factor (ATRX), a component of PML nuclear bodies, as an antiviral factor limiting gene expression through heterochromatinization and plays a role in promoting packaging of viral genomes. Overall, it’s an interesting study that demonstrates multiple roles of ATRX during HCMV replication. The first part of their study investigates the heterochromatinization and gene expression of HCMV when ATRX has been reduced by a knockdown. The authors also show their attempts with cas9 knockout of ATRX, and although they do not go forward because of the variability in the hypomorph cells, it is appreciated that they tried these difficult experiments and included the data for others who are interested in this type of approach. The authors use two strains of HCMV, AD169 and TB40E, to investigate the ability of ATRX to regulate viral IE gene expression and discover similar results except for the total H3 associating with the viral genome which was decreased in TB40E upon ATRX knockdown. In general, the first set of experiments with the gene expression and chromatinization are relatively straightforward and clear, while the investigation of the association of ATRX with viral packaging is a little more difficult to interpret. The one issue is that the DNA fiber assay does not assess if the IdU and CldU is being incorporated into viral DNA or cellular DNA, so it is unclear if the measurements are from viral DNA replication. Interestingly, the ATRX knockdown cells showed significantly less packaged virus, something that will be interesting to explore in the future. This is a great study, the discussion in particular is very well written, and highlights how ATRX may have a dual role as both pro- and anti-viral.

Reviewer #2: This manuscript by Walter et al seeks to probe the mechanism behind the restriction of HCMV growth/life cycle by the cellular protein ATRX. Although ATRX has been known for some time to be part of the cellular intrinsic response to HCMV infection, not very much has been ascertained regarding the actual mechanism of this restriction. Through the use of ATRX knock down cells and several different methods to assess both changes to transcription, DNA chromatinization and replication of the virus, the authors have revealed several important areas where ATRX can affect change in the HCMV infection cycle. The manuscript is clearly written and for the most part, the conclusions drawn are supported by the data as presented. The exception to this comes in their interpretation of the role of ATRX in increasing viral fitness (see below). This reviewer feels this conclusion needs to be supported by a more robust analysis of the actual packaging/secretion of capsids, not just a comparison using titer data. There also needs to be a justification/explanation provided for their use of very low MOI infections into confluent fibroblast monolayers, as this will be a bit counterintuitive to most readers. Please see below for specific comments/suggestions for the authors.

Major points to address:

1. It is clear from the methods section (starting at line 354) that the authors are performing their experiments at very low MOI in completely confluent monolayers. Why have they chosen to perform their experiments under these conditions? Certainly the confluence arrested cells will allow spread of the infection without the problems of replicating cells, but performing experiments at such a low MOI will complicate the interpretation of numbers of cells actually expressing IE/E/L proteins and numbers of cells actually replicating and shedding virions.

2. In Fig 2A and B, the relative levels of IE1/2 mRNA are quite variable within the KD cells. Is this because of varying KD efficiency? If so, this should be stated.

3. In Fig 2D and E, the authors show that there is an “increased level” of all three classes of proteins within the ATRX KD cells at later times pi. Although the quantitation of these blots is reasonable, what would be much more informative here would be whether there were changes in the number of cells positive for each of these proteins, or whether this just represents changes in the level of protein expression in the same number of cells. An immunofluorescent analysis would quickly answer this question. It could be very informative, as this reviewer would guess that there may be, in fact, more cells that show IE positivity in the KD cells which would lead to an increase in E/L protein expression as well.

4. On lines 173-4, the authors point out that the MIEP DE is not very GC rich and yet it becomes more accessible after ATRX KD….any speculation as to why??

5. In Fig 5A and B, the authors see increased levels of viral DNA within ATRX KD cells infected with either virus. Once again, there is a problem with the assessment of “same level of IE1 and UL44” in Fig 5C as measured by Western blot quantitation. Have you looked at these PAA treated cells to ensure that the same number of cells are viral Ag positive prior to release? If there are more Ag positive ATRX KD treated cells, when the block is released, it would make complete sense that more vDNA will accumulate. This needs to be checked, again, using a simple IF experiment.

6. The data presented in Fig 6A-D is problematic. First, the data in Fig 6A posits that on average, 4 genomes are deposited per cell in an MOI=0.1 infection. If this is indeed the case, the number of defective genomes within the cultures is very high. That being the case, Fig 6C indicates that there are roughly twofold more cells that are UL44 foci positive at 48hpi. The question is how many more cells were IE1 positive in the ATRX KD population at 3h, 24h and 48hpi? If the transcription of the MIEP is much more efficient in these KD cells, it would make complete sense that there would be more IE+ cells, which would translate to more UL44 foci+ cells after 48hpi. Again, this needs to be quantitated.

Although there appears to be a statistical difference between the two cell populations with respect to the number of UL44 foci within the cells, it certainly appears that the vast majority of both cell populations have no UL44 foci. It would be very helpful in interpreting this data if a table was shown with how many actual cells had each number of foci. Is the graph to be correctly interpreted as there is only one ATRX KD cell that has 6 foci? If twice as many cells are UL44 positive, one would posit that just by chance there may be cells with more foci within them.

7. In Fig 7, the authors state on line 254 that the “relative percentage of packaged viral genomes was significantly reduced in ATRX KD cells” (as shown in Fig 7B). First, the authors can say nothing about the number of “packaged viral genomes” with the methods they are using, as they are comparing to infectious virions. As is clear from the experiments in Fig 6, these cells produce a large number of defective particles. These could be partial DNA genomes, non-enveloped particles, etc. But there also could be an actual decrease in the numbers of capsids or improper trafficking or envelopment that is taking place in these ATRX KD cells. If the authors wish to speak at all about improper packaging, they would need to perform some TEM experiments to actually look at the nuclei and cytoplasm of these infected cells to analyze the actual ultrastructure of the particles. The authors do not know that the extra genomes are “not packaged”, but only that there is not an increase in the number of infectious progeny produced within these cells.

Reviewer #3: This study by Walter and colleagues investigates the role of the chromatin remodeling factor ATRX for human cytomegalovirus infection. It is well known that ATRX acts as a restriction factor against HCMV and several other herpesviruses and this negative effect is antagonized by the tegument protein pp71 during HCMV infection. So far, it was shown that a knockdown of ATRX increases the number of cells that are able to initiate immediate early gene expression. Using ATRX knockdown fibroblasts, Walter and colleagues perform a broader characterization of the effects of ATRX on HCMV transcription, DNA replication and the release of infectious viruses. They conclude that ATRX acts both in a pro- and an antiviral manner: On the one hand, ATRX appears to increase heterochromatin formation (as deduced from ATACseq and ChIP experiments) resulting in reduced transcription. Interestingly, the authors postulate that not only viral transcription but also viral DNA replication are affected in a negative manner. On the other hand, a knockdown of ATRX results in a disproportionate low increase of released infectious virus. The authors conclude that this is due to a positive effect of ATRX on the packaging of viral genomes. While these are interesting hypotheses, I feel that some of the assumptions are not sufficiently supported by the experimental data of the manuscript. To my opinion (since ATRX affects the onset of IE gene expression), it is not clear whether effects on DNA replication are direct or indirect. For instance, newly replicated DNA was quantitated in PAA arrested/released cells to achieve equal expression of IE1 and UL44. However, this is a rather artificial setup allowing for numerous other reasons why DNA replication is increased in ATRX-KD cells. The DNA fiber assay, which was used to calculate the rate of DNA replication was performed at 48 hpi. However, at this time point, an increase of UL44 (and probably other replication proteins) is already evident in ATRX-KD cells. Thus, the increased rate of DNA replication could simply result from higher levels of the respective replication levels. Furthermore, the authors postulate that ATRX affects the packaging of viral DNA, however, the release of infectious viruses is used as a proxy for that. A more direct experimental setup is necessary in order to conclude on a positive effect of ATRX on the packaging of viral DNA. So in summary, this is an interesting paper describing for the first time that ATRX may act both in an anti- and proviral manner. However, the data set of this paper is (presently) rather limited and some of the conclusion need additionsl experimental support.

**Part II – Major Issues: Key Experiments Required for Acceptance**

Reviewer #1: Figure 6E-I: The DNA fiber assay is difficult, and I appreciate the authors using this to evaluate DNA replication. One concern is whether the DNA that was evaluated in the fiber assay was viral or cellular. The authors do not mention if they used a DNA probe specific for HCMV to localize the incorporated IdU and CldU. Alternatively, do whole cell IFA and co-localize IdU, CldU replication compartments with UL44 to show that the IdU and CldU is incorporating into viral DNA. The other potential way is to isolate the IdU, CldU labeled DNA and perform qPCR to determine the fraction of viral and cellular DNA.

There is no doubt that the PAA is inhibiting DNA replication, but panel F would be more convincing if there was a wider view or even a DAPI stain showing that there is DNA but no IdU, CldU incorporation. An alternative might be to stain whole cells that have been pulsed with IdU and CldU along with DAPI to show that there is no incorporation when treated with PAA.

Figure 2E shows significant increase in the amount of IE1 (6 hpi) and UL44 (24 hpi) protein in the ATRX knockdown compared to control, while Figure 5E shows no significant difference between IE1 and UL44 (48 hpi +PAA) in the ATRX knockdown. Did the authors assess IE1 and UL44 at earlier times following PAA treatment to see if the difference noted in Fig. 2E was present with PAA treatment?

Reviewer #2: please see comments listed above.

Reviewer #3: 1. The authors observe increased IE transcription in ATRX-KD cells. However, it is well known that ATRX can induce a shutdown of HCMV gene expression. Thus, an important question that needs to be answered is, whether the observed increased IE transcript levels are due to an increased number of cells that initiate IE transcription or more IE transcripts per cell. Fig. 2 shows only the quantification of IE transcript levels at 3 hpi, however, transcript quantification should be done for more transcripts and also at later time points of infection. An elegant approach for this would be a single cell RNAseq

2. The authors describe decreased heterochromatin levels in ATRX KD cells and conclude that ATRX induces heterochromatinazation of viral DNA leading to reduced chromatin accessibilty. However, the data set shown to support this conclusion is rather limited. For FAIRE and ChIP only the MIEP was investigated. Only ATAC-seq was performed in a genome wide manner, but the presentation of the data is not clear for me: the authors mention that accessibilty of the distal enhancer is altered but no functional data concerning the consequences of this altered chromatin accessibility are included. Table 1 lists the top 10 loci of differential accessibility but no further explanation concerning the consequences are included.

To my opinion, the authors should include a better and more precise interpretation of ATAC-seq data (does ATRX affect the global heterochromatinazation of HCMV DNA or are there specfic hotspots, is there a consequence of differential accessibility of the distal enhancer region) but should also perform additional experiments (ChIPseq instead of ChIPqPCR, include more chromatin markers).

3. The conclusion that HCMV DNA packaging is affected by ATRX is not well supported by the experimental data. The authors need to think about an experimental approach to more directly measure viral genome packaging (e.g. EM-quantifcation of C-type capsids in ATRX knockdown and control cells).

**Part III – Minor Issues: Editorial and Data Presentation Modifications**

Reviewer #1: Page 2, line 30 (abstract): the word “infectious” should probably be replaced with “infection”.

Page 19, line 412 (materials and methods): It is unclear what the primer sequences are for IE1 and GAPDH used in the qPCR assay.

Reviewer #2: 1. On line 30, the word infectious should be changed to infections.

2. On line 231, the authors reference that cellular DNA synthesis is inhibited in HCMV-infected fibroblasts. It would not be inhibited in actively dividing cells infected at this low an MOI. The authors see no cellular replication because their cells are contact inhibited. This should be corrected.

3. The authors have put together a nice model figure (Fig 8). It is never referred to in the text….

Reviewer #3: The introduction contains several statements that are not correct:

1. Line 67: it is stated that only Daxx deposits histones on HCMV DNA. This is not correct: it was shown that HIRA is also important (PMID: 28981850).

2. In line 66 it is stated that H3K9me3 is repressive. However, there are numerous publications demonstrating that this chromatin mark can also be associated with transcriptional activation and mRNA elongation by RNA pol II. This is important because it challenges the conclusion of the paper that ATRX leads to an increase of heterochromatin.

PLOS authors have the option to publish the peer review history of their article (what does this mean?). If published, this will include your full peer review and any attached files.

Reviewer #1: No

Reviewer #2: No

Reviewer #3: No
---

## [Decision Letter · Decision Letter 1]

20 Aug 2024

Dear Dr. Kalejta,

We are pleased to inform you that your manuscript 'ATRX restricts Human Cytomegalovirus (HCMV) viral DNA replication through heterochromatinization and minimizes unpackaged viral genomes.' has been provisionally accepted for publication in PLOS Pathogens.

Before your manuscript can be formally accepted you may need to complete some formatting changes, which you will receive in a follow up email. A member of our team will be in touch with a set of requests.

Best regards,

Laurie T Krug, PhD

Academic Editor

PLOS Pathogens

Patrick Hearing

Section Editor

PLOS Pathogens

Michael Malim

Editor-in-Chief

PLOS Pathogens

orcid.org/0000-0002-7699-2064

Reviewer Comments (if any, and for reference):

Reviewer's Responses to Questions

**Part I - Summary**

Reviewer #2: As this was a resubmission, I evaluated the changes/additions made by the authors. I commend them for clearly addressing my concerns and for greatly strengthening their manuscript with the addition of several new figures. I have no further suggestions.

Reviewer #3: The authors have addressed all major concerns raised by this reviewer. Importantly, they showed that increased viral IE transcription in ATRX kd cells is not due to a higher number of cells initiating viral gene expression (at least at an MOI of 0.1). Furthermore, the additional experiments performed to narrow the effect of ATRX on the packaging of viral DNA have considerably improved the manuscript.

**Part II – Major Issues: Key Experiments Required for Acceptance**

Reviewer #2: None

Reviewer #3: No further experiments required.

**Part III – Minor Issues: Editorial and Data Presentation Modifications**

Reviewer #2: None

Reviewer #3: No minor issues detected.

PLOS authors have the option to publish the peer review history of their article (what does this mean?). If published, this will include your full peer review and any attached files.

Reviewer #2: No

Reviewer #3: No

---

## [Editor Report · Acceptance letter]

30 Aug 2024

Dear Dr. Kalejta,

We are delighted to inform you that your manuscript, "ATRX restricts Human Cytomegalovirus (HCMV) viral DNA replication through heterochromatinization and minimizes unpackaged viral genomes.," has been formally accepted for publication in PLOS Pathogens.

Best regards,

Michael Malim

Editor-in-Chief

PLOS Pathogens

orcid.org/0000-0002-7699-2064